# Molecular insights into ago-allosteric modulation of the human glucagon-like peptide-1 receptor

Zhaotong Cong[1,2,13], Li-Nan Chen[3,13], Honglei Ma[2,13], Qingtong Zhou[4,13], Xinyu Zou[5], Chenyu Ye[1,2], Antao Dai[6], Qing Liu [6], Wei Huang[7], Xianqiang Sun[7], Xi Wang[2,8], Peiyu Xu [2], Lihua Zhao[2], Tian Xia[5], Wenge Zhong[7], Dehua Yang [2,6,8✉], H. Eric Xu [2,8✉], Yan Zhang [3,9,10,11,12✉] & Ming-Wei Wang [1,2,4,6,8,12✉]

The glucagon-like peptide-1 (GLP-1) receptor is a validated drug target for metabolic disorders. Ago-allosteric modulators are capable of acting both as agonists on their own and as efficacy enhancers of orthosteric ligands. However, the molecular details of ago-allosterism remain elusive. Here, we report three cryo-electron microscopy structures of GLP-1R bound to (i) compound 2 (an ago-allosteric modulator); (ii) compound 2 and GLP-1; and (iii) compound 2 and LY3502970 (a small molecule agonist), all in complex with heterotrimeric $G_s$. The structures reveal that compound 2 is covalently bonded to C347 at the cytoplasmic end of TM6 and triggers its outward movement in cooperation with the ECD whose N terminus penetrates into the GLP-1 binding site. This allows compound 2 to execute positive allosteric modulation through enhancement of both agonist binding and G protein coupling. Our findings offer insights into the structural basis of ago-allosterism at GLP-1R and may aid the design of better therapeutics.

[1] School of Pharmacy, Fudan University, Shanghai, China. [2] The CAS Key Laboratory of Receptor Research, Shanghai Institute of Materia Medica, Chinese Academy of Sciences, Shanghai, China. [3] Department of Biophysics and Department of Pathology of Sir Run Run Shaw Hospital, Zhejiang University School of Medicine, Hangzhou, China. [4] School of Basic Medical Sciences, Fudan University, Shanghai, China. [5] School of Artificial Intelligence and Automation, Huazhong University of Science and Technology, Wuhan, China. [6] The National Center for Drug Screening, Shanghai Institute of Materia Medica, Chinese Academy of Sciences, Shanghai, China. [7] Qilu Regor Therapeutics, Inc., Shanghai, China. [8] University of Chinese Academy of Sciences, Beijing, China. [9] MOE Frontier Science Center for Brain Research and Brain–Machine Integration, Zhejiang University School of Medicine, Hangzhou, China. [10] Key Laboratory of Immunity and Inflammatory Diseases of Zhejiang Province, Hangzhou, China. [11] Zhejiang Laboratory for Systems and Precision Medicine, Zhejiang University Medical Center, Hangzhou, China. [12] School of Life Science and Technology, ShanghaiTech University, Shanghai, China. [13]These authors contributed equally: Zhaotong Cong, Li-Nan Chen, Honglei Ma, Qingtong Zhou. ✉email: dhyang@simm.ac.cn; eric.xu@simm.ac.cn; zhang_yan@zju.edu.cn; mwwang@simm.ac.cn

As one of the most well-known class B1 G-protein-coupled receptors (GPCRs), the glucagon-like peptide-1 receptor (GLP-1R) is a clinically validated drug target for type 2 diabetes and obesity[1–3]. Despite the therapeutic success, currently available GLP-1R agonists are suboptimal due to poor patient compliance caused by subcutaneous injections and several side effects such as nausea and vomiting[4,5]. Oral delivery of GLP-1 mimetics has been pursued for many years, with semaglutide being the first and only one in the clinic albeit its low bioavailability and frequent gastrointestinal complaints[6,7]. Thus, the development of orally active small molecule GLP-1R modulators remains an attractive strategy and a few compounds have progressed to clinical trials[8].

The scientific advances in GPCR structural biology, fueled by tremendous interests from both academia and industry, led to 16 solved GLP-1R structures. In 2017, two inactive structures bound to negative allosteric modulators (NAMs)[9], one intermediate structure bound to a truncated peptide agonist[10], and one active structure in complex with GLP-1 and heterotrimeric $G_s$[11] were reported. This was followed by eight G protein-bound structures in complex with peptides (GLP-1 and exendin-P5)[12,13], non-peptidic agonists (TT-OAD2, PF-06882961, CHU-128, RGT1383, and LY3502970)[13–16], and a positive allosteric modulator (PAM) LSN3160440[17], as well as a peptide-free *apo* state[18] and three thermal-stabilized transmembrane domain (TMD)[9] structures. They display diversified conformations and reveal key information on ligand recognition and GLP-1R activation. Integrated with the functional, pharmacological, and computational data, these studies demonstrate distinct locations and different conformations of the orthosteric binding pocket between small molecule and peptidic ligands.

Allosteric modulation by PAMs that bind anywhere distinct from the orthosteric site of endogenous ligands and enhance the activity of agonists in an uncompetitive way is an alternative approach to peptide therapy[19–22]. Previous studies showed that several PAMs of GLP-1R, such as the most characterized electrophilic chemotype compound BETP[23,24] and the substituted quinoxaline compound 2 (6,7-dichloro-3-methanesulfonyl-2-tert-butylamino-quinoxaline)[25,26], were able to initiate or promote receptor activation[9,27]. Apart from increasing the binding affinity of GLP-1, compound 2 also acts as an agonist on its own, thereby being classified as an ago-allosteric modulator (ago-PAM) of GLP-1R[25,28]. Specifically, in the absence of an orthosteric ligand, compound 2 displayed a strong partial agonism in cAMP response (80% $E_{max}$ of GLP-1), a weak partial agonism in pERK1/2, and no detectable calcium mobilization response[29,30]. It also caused less GLP-1R internalization than GLP-1, due to the weaker β-arrestin recruitment (30–50% $E_{max}$ of GLP-1)[30]. As a PAM, compound 2 caused a concentration-dependent increase in the affinity of GLP-1 and oxyntomodulin[31] and induced biased signaling in a probe-dependent manner[31,32]. In addition, compound 2 was also reported to modulate cAMP responses elicited by small molecule agonists including Boc5, BMS21, and TT15[29]. Although structurally and functionally unique, this compound has poor pharmacokinetic properties due to its electrophilic nature that preclude potential clinical development[33]. Interestingly, like BETP, compound 2 interacts with cysteine 347[6.36] via covalent modification at the cytoplasmic side of GLP-1R, consistent with our previous observation that an intact extracellular domain (ECD) is required for the activity of BETP and compound 2[34] albeit the basis for such a requirement remains unknown. This distinct feature prompts us to use compound 2 as a tool to study the structural basis of its ago-allosterism in the context of both peptidic and non-peptidic ligands.

Here we report high-resolution cryo-electron microscopy (cryo-EM) structures of human GLP-1R–$G_s$ in complex with compound 2 alone, with both compound 2 and endogenous peptidic agonist GLP-1, or with compound 2 and a small molecule agonist (LY3502970). The structural information obtained from this study provides valuable insights into molecular mechanisms by which compound 2 exerts the ago-allosteric action and its relevance to drug discovery.

## Results

**Structure determination**. To understand activation and allosterism of compound 2 on GLP-1R, we determined the structures of GLP-1R–$G_s$ complexes bound to compound 2 alone and in the presence of GLP-1 or LY3502970 (Fig. 1). The NanoBiT tethering strategy and Nb35 were used to stabilize the protein complexes[35,36] (Supplementary Fig. 1a). These protein complexes were then purified, resolved as monodispersed peaks on size-exclusion chromatography (SEC), and verified by SDS gel and negative staining to ascertain all the expected components (Supplementary Fig. 1c–e). Vitrified complexes were imaged by cryo-EM. After sorting by constitutive 2D and 3D classifications, 3D consensus density maps were reconstructed with global resolutions of 2.5 Å (without ECD) or 3.3 Å (with ECD) for compound 2 alone, 2.5 Å for compound 2 plus GLP-1, and 2.9 Å for compound 2 plus LY3502970, respectively (Supplementary Fig. 2 and Supplementary Table 1). The cryo-EM maps allowed us to build an unambiguous model for most regions of the complexes except for the flexible α-helical domain (AHD) of Gα and the stalk between transmembrane helices 1 (TM1) and ECD, which were poorly resolved in most cryo-EM structures of GPCR–$G_s$ complex (Supplementary Fig. 3). In addition, residues N338 to T343 in the intracellular loop 3 (ICL3) of the compound 2–GLP-1–GLP-1R–$G_s$ and compound 2–LY3502970–GLP-1R–$G_s$ complexes, F369 to R376 in the extracellular loop 3 (ECL3) of the compound 2–GLP-1R–$G_s$ and compound 2–LY3502970–GLP-1R–$G_s$ complexes, and P56 to F61 in the ECD of the compound 2–GLP-1R–$G_s$ complex were also poorly resolved and thus omitted from the corresponding final model. The ECD in the GLP-1 and compound 2-bound complexes was clear enough to enable modeling of the backbone and a majority of side-chains using the ECD and TMD-refined maps (Fig. 1).

**Binding of compound 2**. The compound 2–GLP-1–GLP-1R–$G_s$ complex has a typical active assembly close to that of GLP-1–GLP-1R–$G_s$ with Cα root mean square deviation (RMSD) of 0.71 Å for the whole complex (Fig. 2a). Compared with the peptide-free *apo* and intermediate state structures, compound 2 rendered the TM bundle and the ECD of GLP-1R undergo extensive conformational transitions (Supplementary Figs. 4 and 5). Specifically, starting from a closed conformation in *apo* state, the ECD of GLP-1R bound by either compound 2 or truncated peptide agonist (peptide 5)[10] rotated clockwise and approached ECL1 and ECL2. The extracellular parts of TM1 and TM2 moved inward to the center, while TM7 moved outwards accompanied by conformational changes of ECL1. In the intracellular domain, the sharp kink around the conserved Pro[6.47b]-X-X-Gly[6.50b] motif in TM6 pivoted the intracellular half of TM6 to move outwards by 18.4 Å (measured by Cα carbon of K346[6.35b]), to the same extent as that achieved by orthosteric agonists (Supplementary Fig. 4). This was joined by a modest outward movement of TM5 (3.6 Å measured by Cα carbon of K336[5.66b]), thereby creating an intracellular crevice to accommodate $G_s$ coupling[11,12,18]. The latter was anchored by the α5 helix of Gα$_s$ (GαH5) and formed rich contacts with TMs 2-7 and ICLs 1-3 (Supplementary Fig. 6).

Instead of occupying the orthosteric binding pocket where peptide or small molecule agonists generally bind, the high-

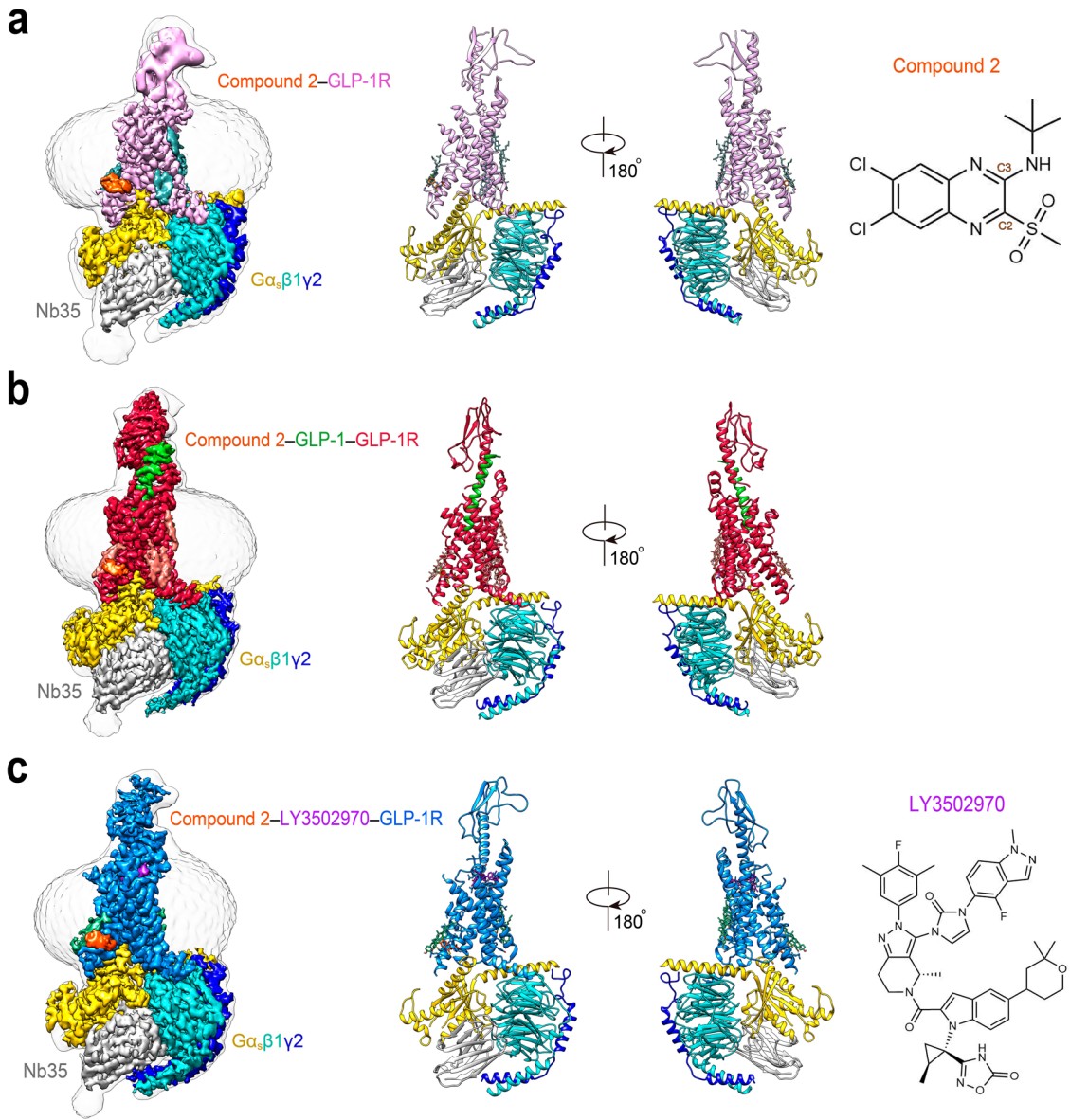

**Fig. 1 The overall cryo-EM structure of GLP-1R–G$_s$ complexes.** The cryo-EM maps with a disc-shaped micelle (left) and cartoon representation (middle) of compound 2-bound complex (**a**), compound 2 and GLP-1-bound complex (**b**), and compound 2 and LY3502970-bound complex (**c**). The chemical structures of compound 2 and LY3502970 are shown in the right panel of **a** and **c**. Compound 2 and GLP-1-bound GLP-1R in red; compound 2-bound GLP-1R in hot pink; compound 2 and LY3502970-bound GLP-1R in blue; Gα$_s$ Ras-like domain in yellow; Gβ subunit in cyan; Gγ subunit in navy blue; Nb35 in gray; GLP-1 in green; compound 2 in orange; LY3502970 in purple.

resolution cryo-EM map demonstrates that compound 2 is covalently bonded to C347[6.36b] (Wootten numbering in superscript[37]) and mounted on the membrane-facing surface of the cytoplasmic end of TM6, providing solid structural evidence of a unique binding site for the ago-allosteric modulator (Fig. 2b and Supplementary Fig. 3d). Such a covalent modification is supported by a previous report showing that non-sulfonic substituents at the C-2 position (methylsulfone) failed to produce measurable cAMP responses[38], consistent with our mutagenesis results, where C347[6.36b]A mutation diminished the potency of compound 2 without affecting that of GLP-1 (Fig. 2c and Supplementary Table 2). Compound 2 forms predominantly hydrophobic interactions with the adjacent residues in TM6. The tert-butyl moiety of compound 2 points to TM7 and makes hydrophobic contacts with A350[6.39b] and K351[6.40b], replacement at the C-3 position by polar functional groups caused a dramatic decline in its potency and efficacy[38]. The dichloroquinoxaline

group extends to ICL3 forming nonpolar interactions with K346[6.35b], C347[6.36b], and a cholesterol molecule in TM6, the introduction of electron-donating substituents or replacement of quinoxalines by benzimidazole or quinoline at C-6 and C-7 positions led to poor tolerance[38]. The bulky A350[6.39b]W and K351[6.40b]A mutants almost abolished the maximal response ($E_{max}$) of GLP-1R-mediated cAMP accumulation in presence of compound 2, while V332[5.62b]A, K346[6.35b]A, and L349[6.38b]A mutants greatly elevated basal cAMP activities but significantly diminished the efficacy of the response upon stimulation by either compound 2 or GLP-1, suggesting these mutations affect the kink of TM6 required for receptor activation. The same covalent mechanism of action was previously demonstrated for another PAM of GLP-1R (i.e., BETP), showing that in addition to C347, BETP also formed a covalent adduct with C438 in the C terminus[27] that is invisible in the current GLP-1R complex structures. Although mutation of C438 did not alter the PAM

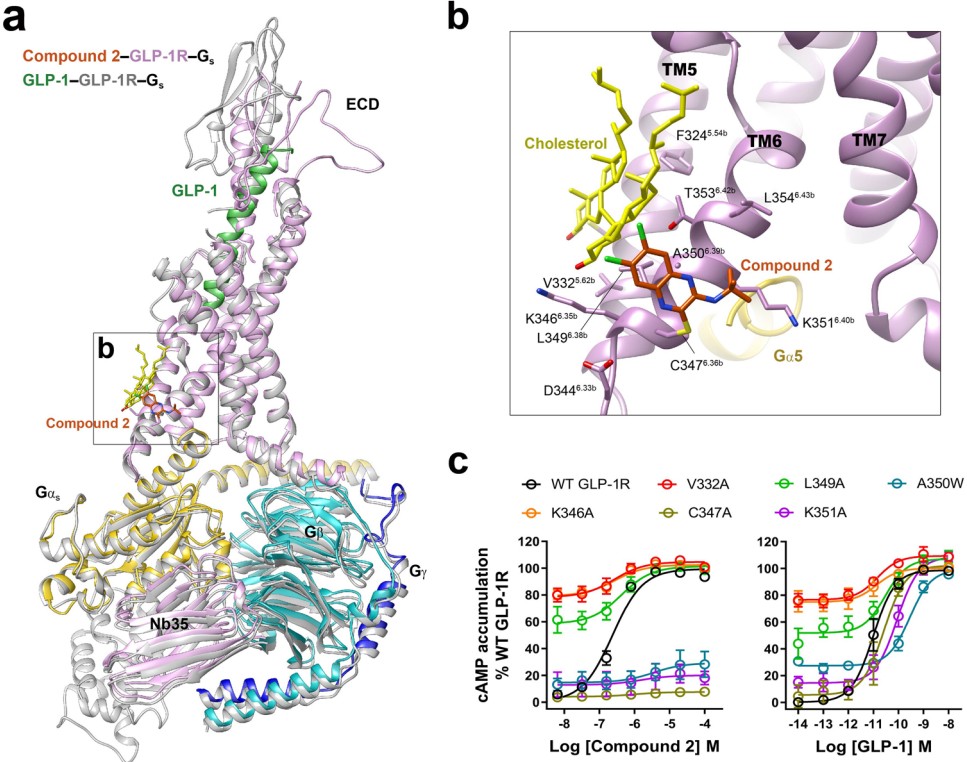

**Fig. 2 Unique agonist-binding site in GLP-1R for compound 2. a** Superimposition of the active state GLP-1R bound by compound 2 or GLP-1 (PDB code: 6X18) reveals a high structural similarity except for the ECD region. **b** Compound 2 covalently binds to C347[6.36b] and mounts on the membrane-facing surface of the cytoplasmic end of TM6. **c** Representative mutation effects on the ago-allosterism associated with compound 2. Data are presented as means ± S.E.M. of three independent experiments. WT, wild-type.

activity of BETP or compound 2[27], it cannot rule out the possible formation of a covalent adduct at C438 in our compound 2-bound GLP-1R structure.

**ECD conformation.** The most profound structural feature in the extracellular half of compound 2-bound GLP-1R is the unique position and orientation of ECD, distinct from all available full-length GLP-1R structures reported to date (Fig. 3). The tripartite α-β-β/α architecture of ECD observed in *apo* or peptide-bound structures[39] was partially disturbed in the compound 2–GLP-1R–G[s] complex except for the N-terminal α-helix (residues 29 to 49). GLP-1-bound ECD was shown to be fully extended[11,34] whereas that of compound 2-bound folded down towards the TMD core and penetrated into the orthosteric binding pocket through its N-terminal α-helix and loop by inward movement of 9.0 Å (measured at the Cα of R40). Notably, the orientation of ECD is distinct from that of GLP-1-bound GLP-1R with a rotation angle of 97.4 degrees: the former pointed from ECL2 to TM1-TM2 and the latter oriented from ECL1 to TM5-TM6 (Fig. 3a). The tip of the N-terminal α-helix (T29) moved by 19.9 Å and inserted into a cleft between TM1 and TM2, partially overlapping with the recently reported allosteric site of PAM LSN3160440[17]. Locked by the conserved disulfide bond (C46-C71) with the N-terminal α-helix, β1 strand (residues 61 to 77) of GLP-1R ECD also shifted towards ECL1 by 9.3 Å (measured at the Cα of E68). Such a movement consequently shortened the length of α-helix at ECL1 in *apo* state and extended TM2 by one turn, thereby stabilizing the ECD confirmation via an extensive network of complementary polar and non-polar contacts (Fig. 3a and Supplementary Fig. 5). Consistently, our molecular dynamics (MD) simulations found that the ECD interacted with the TMD intimately, such that the

N-terminal α-helix stably inserted to the TMD core (Supplementary Fig. 7).

To reveal functional roles of the ECD in the presence of either compound 2 or GLP-1, we truncated the N-terminal α-helix in a systemic manner and measured cAMP responses subsequently (Fig. 3b and Supplementary Table 2). For compound 2, truncation of a large portion (28–48 residues) of the N-terminal α-helix resulted in a dramatic efficacy decrease of compound 2, implying the role of the tip of N-terminal α-helix in the ago-allosterism. Removal of the N-terminal 55 residues or ECD deletion completely abolished the activation effect of compound 2 despite that it binds to the cytoplasmic side of TM6, consistent with our previous report that ECD is required for GLP-1R and GCGR activation[34]. Besides, similar dose-response characteristics between GLP-1 and compound 2 suggest that the latter is capable of stimulating GLP-1R independently (ago-allosteric activation). In contrast, the action of GLP-1 is dependent on the ECD which binds to the C-terminal half of the peptide. Truncation of the ECD by 28 or 33 residues reduced GLP-1 potency by 50-fold and 13-fold, respectively. In line with the previous study[34], cAMP signaling was completely abolished when 38 residues were truncated, but this shortened construct still worked for compound 2. This phenomenon was also observed in constructs expressing wild-type and mutant glucagon receptors. For instance, F345[6.36b]C mutant of GCGR was sufficient to confer sensitivity to compound 2 (Fig. 3b and Supplementary Table 3). These results show that the ECD requirement is ligand-dependent and is differentiated between orthosteric and allosteric modulators.

To further address this point, we analyzed the TMD-ECD interacting patterns by measuring the buried surface area between TMD and ECD or between TMD and agonists for all available GLP-1R structures with visible ECDs (Fig. 3c, d). In the *apo* state, its ECD

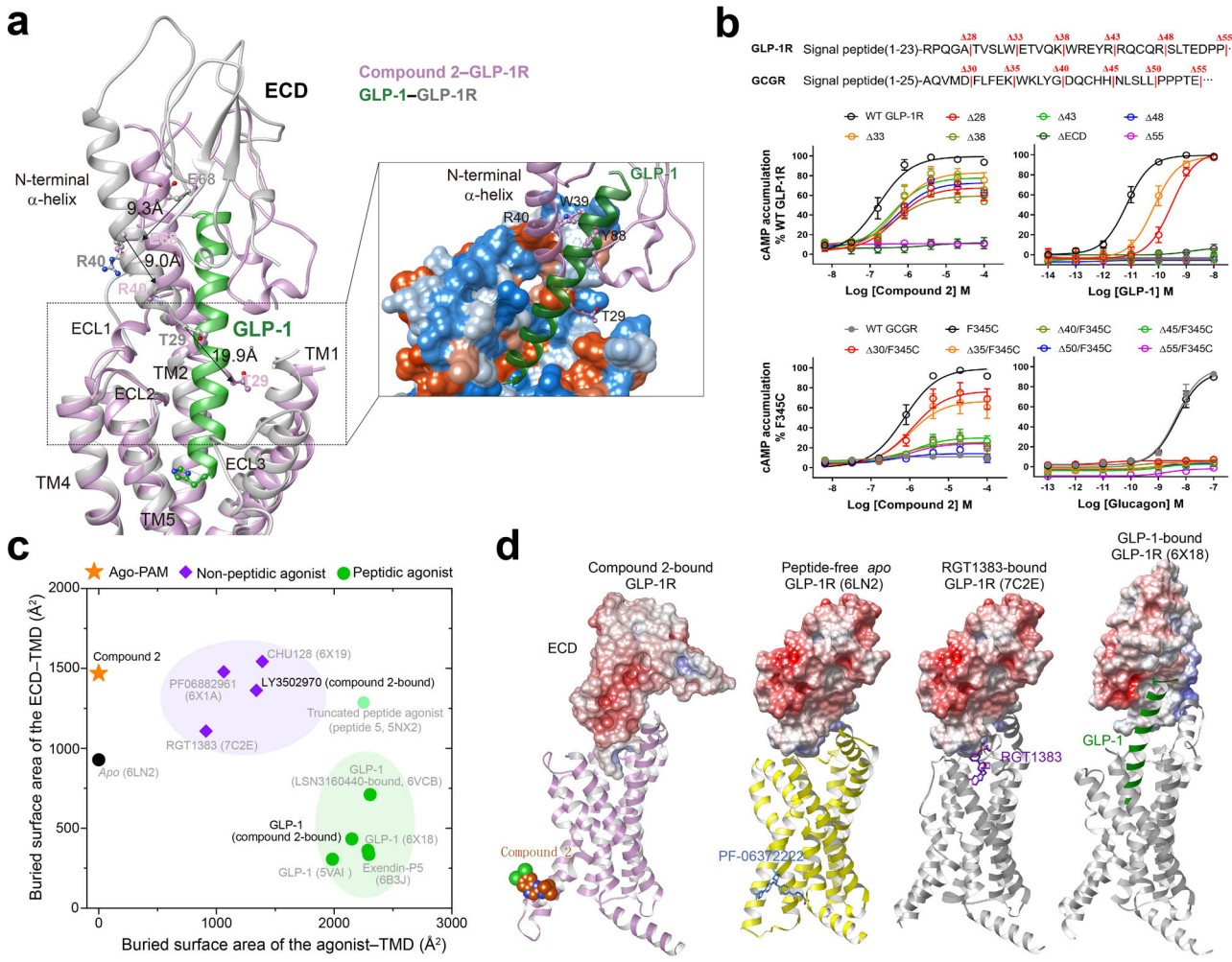

**Fig. 3 Diversified ECD engagements in GLP-1R activation. a** Comparison of the ECD conformation between the GLP-1-bound (PDB code: 6X18) and compound 2-bound GLP-1R. A close-up view of the interaction between ECD and TMD shows that the N-terminal α-helix of ECD penetrated into the orthosteric TMD pocket by a distinct orientation. **b** Signaling profiles of ECD-truncated GLP-1R and GCGR in response to their cognate peptides (GLP-1 and glucagon) and compound 2. Compound 2 is verified as an agonist for GCGR mutant F345$^{6.36b}$C. Data are presented as means±S.E.M. of three independent experiments. WT, wild-type. Δ, residue truncation. **c** Scatter plot of TMD-ECD/agonist interaction of all available GLP-1R structures with visible ECDs. $X$ and $Y$ axis represents the buried interface area of agonist-TMD and ECD-TMD, respectively. The interface areas were calculated using freeSASA. **d** Surface representation of diversified conformations of ECD in representative GLP-1R structures.

adopted a closed TMD-interacting confirmation with a buried surface area of 928 Å$^2$. When bound by peptide agonists such as GLP-1 and exendin-P5, their ECDs stood up along the α-helical peptide and made extensive contacts with the C-terminal half of the peptide, thereby limiting the contact with TMD (~350 Å$^2$). Meanwhile, the N-terminal half of the peptide inserted into the TMD core with massive contacts (buried surface area of more than 2000 Å$^2$). As a comparison, small molecules showed much reduced TMD-interacting surface area (~1000 Å$^2$). Complimentarily, its ECD folded down towards the TMD to stabilize the complex with a buried surface area of ~1500 Å$^2$, remarkably higher than that of peptidic ligands. In the case of compound 2-bound structure, the N-terminal α-helix of ECD penetrated to the TMD core and formed extensive contacts with residues in TM1-3, ECL1, and ECL2, resulting in an ECD-TMD interacting surface area of 1438.4 Å$^2$, similar to other small molecule agonists. The dynamic nature of ECD observed in this study is consistent with previous findings that GLP-1R is equipped with high versatility to recognize a wide range of ligands with distinct chemotypes[40,41] and to participate in diversified receptor activation processes[13,34] (Supplementary Fig. 4).

**Receptor activation.** In spite of various conformational changes upon ligand binding, signaling initiation by either peptidic or small molecule agonists shares a common pathway, i.e., reorganization of the central polar network, HETX motif, and TM2-6-7-helix 8 polar network, as well as the hallmark outward movement of TM6[11,13,42]. Thus, the complexes of GLP-1–GLP-1R–G$_s$ and compound 2–GLP-1R–G$_s$ displayed almost identical conformation reflecting the above polar network rearrangement and TM6 movement (Fig. 4). However, there are some distinct structural features in compound 2 alone bound receptor. At the bottom of the orthosteric pocket, the side chain of R310$^{6.40b}$ pointed to the TMD core and formed a cation-pi stacking with Y241$^{3.44b}$, where it pointed to ECL3 and made no contact with the TMD core in the GLP-1 bound structure. Another notable difference resides in H363$^{6.52b}$, which is adjacent to the conserved Pro$^{6.47b}$-X-X-Gly$^{6.50b}$ motif that pivots the intracellular half of TM6 to move outwards. Among all the active GLP-1R structures, H363$^{6.52b}$ dipped into the TMD core whereas it is reoriented ~90° to an outside-facing position and formed a hydrogen bond with Q394$^{7.49b}$ through dismissal of the hydrogen bond with the

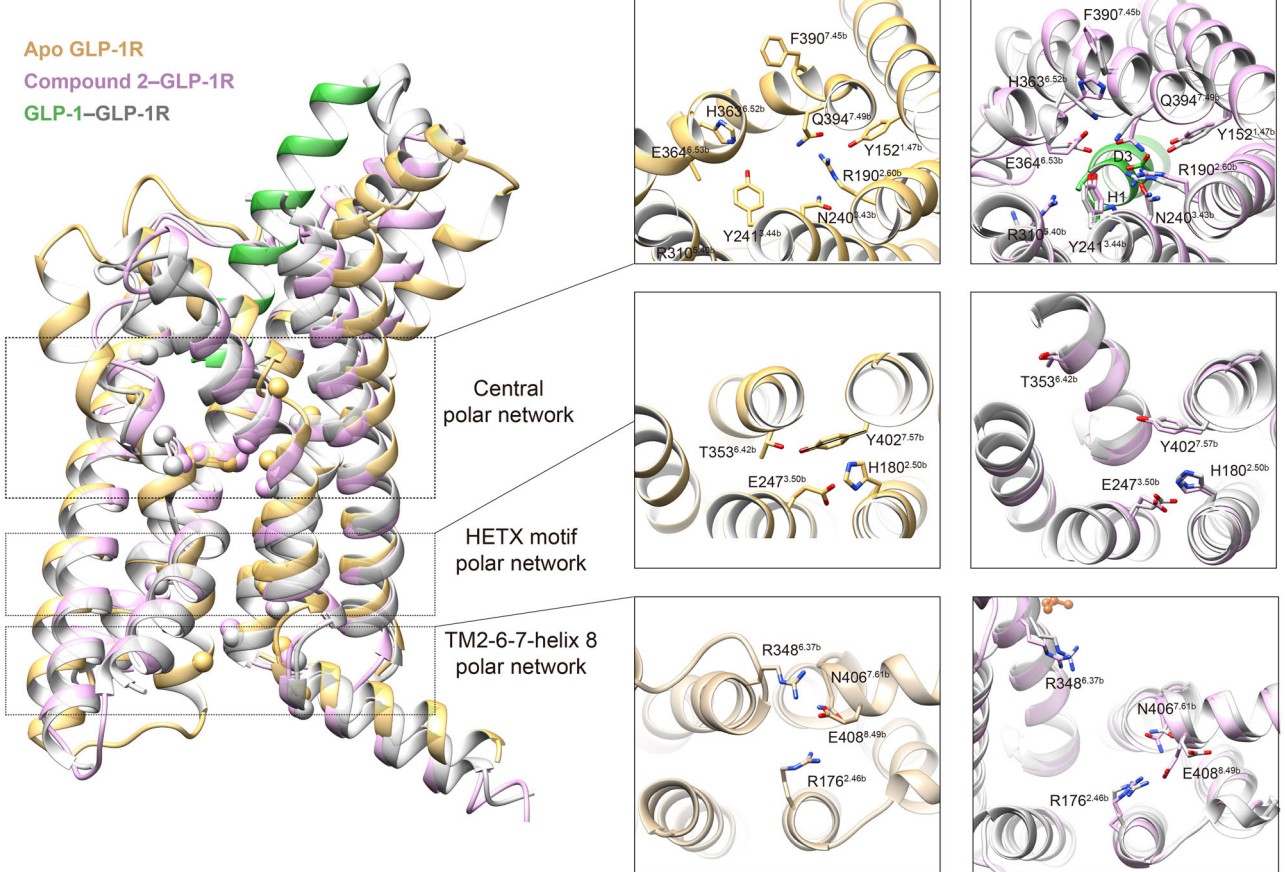

**Fig. 4 Comparison of the polar network rearrangement upon GLP-1R activation triggered by compound 2 and GLP-1.** Left, superimposition of compound 2 bound GLP-1R with the peptide-free *apo* state (PDB code: 6LN2) and GLP-1 bound active state (PDB code: 6X18). Upon receptor activation, three layers of the polar network (central polar network located on the bottom of the orthosteric binding pocket, family-wide conserved HTEX motif polar network, and TM2-6-7-helix 8 polar network that topologically close to G protein) were reorganized. The Cα atoms of the residues that participated in the polar network rearrangements upon receptor activation are shown as spheres. Right, comparison of residual conformation between *apo* state, GLP-1-bound, and compound 2-bound active states for three layers of the polar network, whose residues are shown as a stick with Wootten numbering in superscript.

backbone carbonyl oxygen atom of P358[6.47b]. The distinct features manifested by R310[6.40b] and H363[6.52b] are likely to be responsible for the kink formation induced by compound 2, in line with previous mutagenesis results that mutation at these two sites altered GLP-1R signaling profiles (cAMP, pERK and iCa$^{2+}$)[4,37].

As expected, the compound 2–GLP-1R–G$_s$ complex exhibited a remarkable similarity to GLP-1-bound structures in terms of G protein-binding interface, consistent with a common mechanism of G$_s$ protein engagement. Nonetheless, one additional contact was found to be unique for compound 2-bound structure, i.e., a hydrogen bond between the backbone atom of L260 in ICL2 and R38 in the GαH5 subunit (Supplementary Fig. 6).

**Positive allosterism.** In line with previous findings[25], we found that the potency of GLP-1was not changed but its binding affinity was enhanced by compound 2. The same phenomenon was observed for LY3502970 (Supplementary Fig. 8a). The high-resolution cryo-EM maps of the compound 2–GLP-1–GLP-1R–G$_s$ and compound 2–LY3502970–GLP-1R–G$_s$ complexes provide a good opportunity to further investigate compound 2-associated positive allosterism[25]. As shown in Fig. 5a, the overall structure of the compound 2–GLP-1–GLP-1R–G$_s$ resembles that without this ago-PAM, with a Cα RMSD of 1.0 Å. This observation also applies to LY3502970–GLP-1R–G$_s$ with or without compound 2 (Cα RMSD of 0.8 Å). The binding poses of compound 2 in these two structures are nearly identical to that of

compound 2 alone except for a slightly different orientation (Fig. 5b and Supplementary Fig. 3d). As seen in the compound 2–GLP-1R–G$_s$ complex, two cholesterol molecules were also found in the identical position of the compound 2–GLP-1–GLP-1R–G$_s$ and compound 2–LY3502970–GLP-1R–G$_s$ complexes, implying a common role for the two cholesterol molecules. In the presence of compound 2, both GLP-1 and the ECD moved inwards to the TMD core by 0.7 and 1.3 Å (measured at the Cα of Y19 in GLP-1 and P90 in ECD), respectively. Upon binding of compound 2, several distinct contacts were formed including a hydrogen bond between T11 in GLP-1 and D372 in ECL3, as well as a salt bridge between R36 in GLP-1 and D215 in ECL1 (Supplementary Fig. 8b). Meanwhile, G$_s$ moved upward by 0.5–0.8 Å, especially the αN-helix in Gα$_s$ and Gβ subunits (Fig. 5c). Such an alteration may strengthen G protein coupling by introducing several newly formed polar interactions such as two salt bridges (R176[2.46b] and E408[8.49b], E423[8.64b] and R46 in Gβ). A similar phenomenon was also observed in the compound 2–LY3502970–GLP-1R–G$_s$ structure (Fig. 5c) and received the support of our MD studies in which compound 2-bound GLP-1R could stably bind to GLP-1 in the absence of G protein (Supplementary Fig. 9).

**Discussion**

It has been widely accepted that class B1 GPCRs adopt a two-step model for ligand binding and receptor activation[34,43]. However,

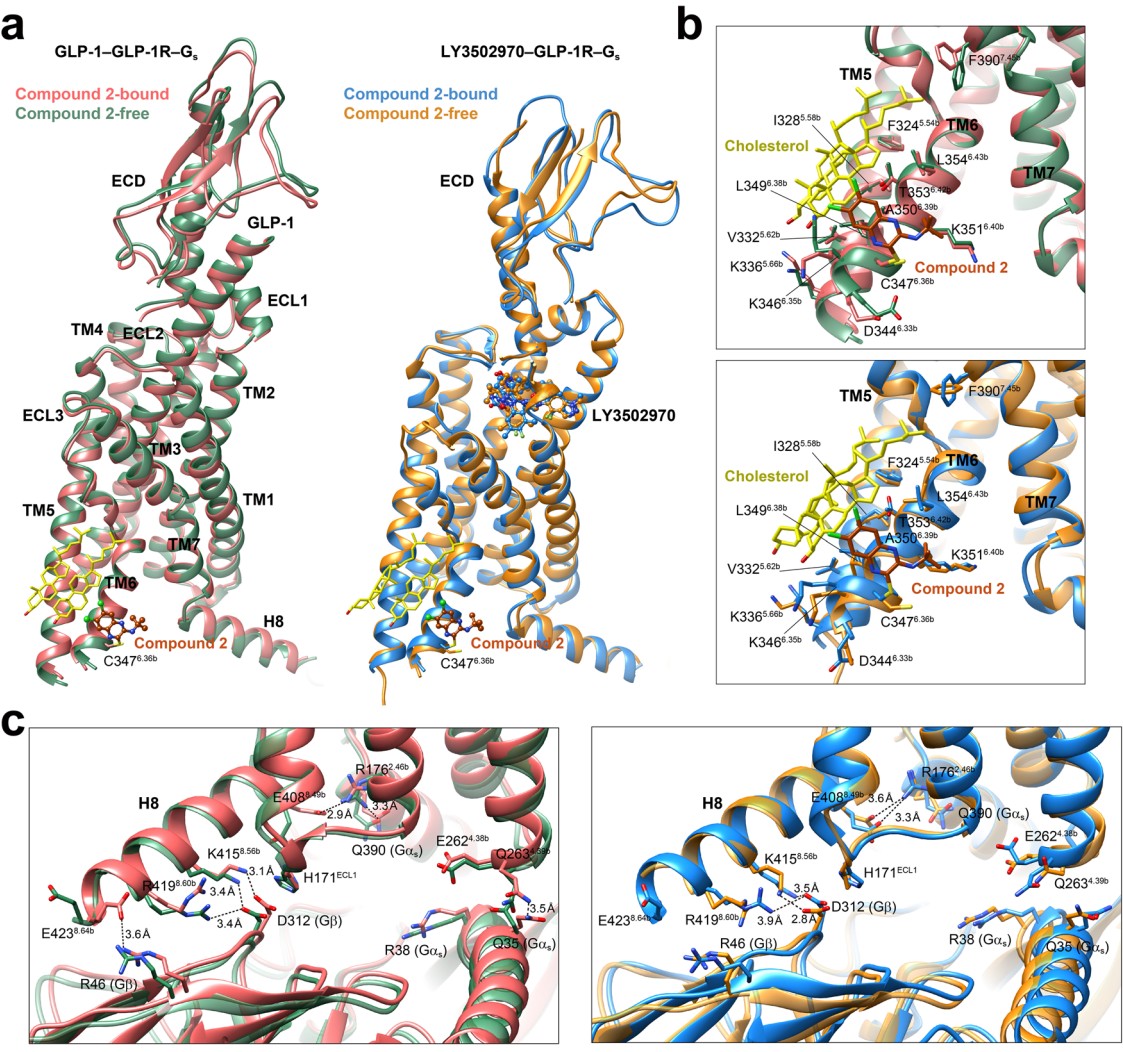

**Fig. 5 Structural insights into the ago-allosterism associated with compound 2. a** Superimposition of the GLP-1–GLP-1R–G$_s$ (left, PDB code: 6X18) and LY3502970–GLP-1R–G$_s$ (right, PDB code: 6XOX) structures with that bound to compound 2. G protein is omitted for clarity. **b** Binding poses of compound 2. Two cholesterols bound to the TM5-TM6 cleft contribute hydrophobic interaction with compound 2. **c** Comparison of the G protein-coupling interfaces of GLP-1 bound (left) and LY3502970 bound (right) active structures with that bound to compound 2.

this may not relevant to small molecule modulators that have different requirements for the ECD. Several recently determined non-peptidic ligand-bound GLP-1R structures[13,14,16] have uncovered previously unknown binding pockets that may have implications in their pharmacological profiles such as biased signaling and allosteric agonism (Supplementary Fig. 10). Ago-PAMs, acting as agonists on their own and as enhancers for orthosteric ligands, are capable of increasing agonist potency and providing additional efficacy[44], thereby offering an alternative to conventional therapeutics. Mounted on the membrane-facing surface of the cytoplasmic tip of TM6, compound 2 was able to induce a newly discovered ECD conformation allowing its N-terminal helix to penetrate into the TMD core, a key step for the ago-allosterism observed experimentally, consistent with the intrinsic agonism hypothesis for the ECD of GLP-1R[40]. Interestingly, despite the extracellular conformational differences induced by compound 2 and orthosteric agonists, their intracellular architectures underwent a similar reorganization of polar residues that enabled the formation of TM6 sharp kink with the help of G protein binding[42,45], indicative of GLP-1R activation (Fig. 6).

Diversified ECD conformations represent the dynamic nature of the receptor in response to a variety of external stimuli. In the

peptide-free *apo* state, the ECD is in favor of a TMD closed conformation stabilized by ECL1 and ECL3[18]. Ligand binding triggers dissociation of the ECD from the TMD, allowing the peptide N-terminus to insert into the orthosteric binding pocket for receptor activation. Unlike peptide, the binding of compound 2 and other small molecule modulators require a complementary conformation of the ECD for ligand recognition, such that the N-terminal α-helix of the compound 2–GLP-1R–G$_s$ deeply inserted to the orthosteric binding pocket as opposed to other GLP-1R structures (Fig. 6), suggesting its unique activation mechanism. In line with this view, we found that the ECD-TMD interaction is crucial for compound 2 and GLP-1 to function. However, this interaction was not required for small molecule agonist TT-OAD2 that elicited cAMP signaling without the presence of an ECD[14]. Reduction in cAMP responses to compound 2 following truncation of the N-terminal helix indicates that the ability of compound 2 to engender a stable ECD conformation may contribute to the "ago" effect on cAMP production. In line with the cryo-EM structure of compound 2-bound GLP-1R, the MD simulations indicate that the N-terminal helix consistently inserts to the TMD core thereby stabilizing the interaction between the ECD and ECL1. This implies that the insertion of the ECD N-

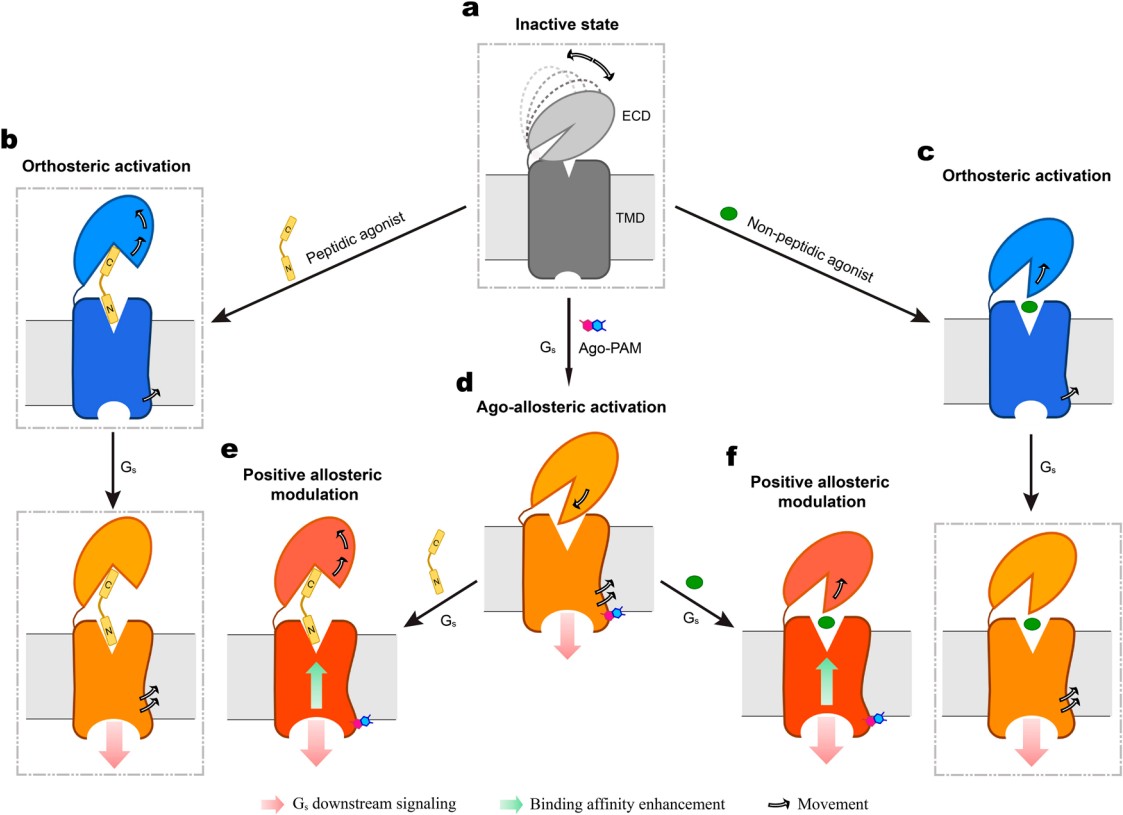

**Fig. 6 Postulated model of GLP-1R activation. a**, In the absence of a ligand, the ECD of GLP-1R is dynamic with multiple conformations (dashed lines) but favors a closed state. The binding of peptidic (**b**) or non-peptidic (**c**) agonist triggers the ECD to disengage from the TMD thereby lowering the energy barrier of G protein coupling (*orthosteric activation*). **d** Ago-PAM, such as compound 2, covalently binds to the cytoplasmic side of TM6 and induces its outward movement. Meanwhile, the ECD bends downwards to the TMD core thereby penetrating into the orthosteric pocket via the N-terminal α-helix. The TMD rearrangement, together with $G_s$ binding, elicits downstream signaling (*ago-allosteric activation*). Ago-PAM also enhances both peptidic (**e**) and non-peptidic (**f**) agonist binding affinity (*positive allosteric modulation*). Inactive, intermediate (agonist-bound), and active (both agonist and G protein-bound) conformations are colored in gray, blue, and orange/orange-red, respectively. Movements of the ECD and the intracellular half of TM6 are indicated by arrows. Cell membranes are shown as gray bilayers. Previously reported GLP-1R structures are indicated by dashed-line boxes.

terminal helix to the TMD core is a unique feature of compound 2-bound GLP-1R. Therefore, the ECD of GLP-1R plays a pivotal role in stabilizing receptor conformation and facilitating its activation as previously proposed[40,46].

Collectively, our structures reveal an ago-PAM mechanism for GLP-1R, in which the rearrangement of the conserved polar network is triggered by TM6 movement and stabilized by the ECD-TMD interaction. This work expands our understanding of GLP-1R activation and provides valuable insights into the potential application of ago-allosterism in drug discovery.

## Methods

**Cell culture**. *Spodoptera frugiperda* (*Sf*9) insect cells (Expression Systems) were grown in ESF 921 serum-free medium (Expression Systems) at 27 °C and 120 rpm. HEK 293T cells (Cell Bank at the Chinese Academy of Sciences) were cultured in Dulbecco's modified Eagle's medium (DMEM, Life Technologies) supplemented with 10% fetal bovine serum (FBS, Gibco) and maintained in a humidified chamber with 5% $CO_2$ at 37 °C. CHO-K1 (ATCC #CCL-61) cells were maintained in F12 containing 10% (v/v) FBS, 100 units/mL penicillin and 100 mg/mL streptomycin at 37 °C in 5% $CO_2$.

**Constructs**. The human GLP-1R was modified with its native signal sequence (M1-P23) replaced by the haemagglutinin (HA) signal peptide to facilitate receptor expression. To obtain a GLP-1R–$G_s$ complex with good homogeneity and stability, we used the NanoBiT tethering strategy, in which the C-terminus of rat Gβ1 was linked to HiBiT subunit with a 15-amino acid polypeptide (GSSGGGGSGGGGSSG) linker and the C-terminus of GLP-1R was directly attached to LgBiT subunit followed by a TEV protease cleavage site and a double MBP tag (Supplementary Fig. 1a). A dominant-negative human Gαs (DNGαs) with

8 mutations (S54N, G226A, E268A, N271K, K274D, R280K, T284D, and I285T) was generated by site-directed mutagenesis to limit G protein dissociation[47]. An engineered $G_s$ construct (mini-$G_s$) was used for expression and purification of the compound 2−LY3502970−GLP-1R complex, which was designed based on that used in the determination of A2AR-mini-$G_s$ crystal structure[48]. The replacement of the original Gαs α-helical domain (AHD, V65-L203) with that of human Gi1 (G60-K180) provided the binding site for Fab_G50, an antibody fragment used to stabilize the rhodopsin-Gi complex[49]. In addition, the substitution of N-terminal histidine tag (His6) and TEV protease cleavage site with the N-terminal eighteen amino acids (M1-M18) of human Gi1 made this chimeric $G_s$ capable of binding to scFv16 used to stabilize GPCR-Gi or GPCR-G11 complexes[50,51]. The constructs were cloned into both pcDNA3.1 and pFastBac vectors for functional assays in mammalian cells and protein expression in insect cells, respectively. All modifications of the receptor had no effect on ligand binding and receptor activation (Supplementary Fig. 1b and Supplementary Table 2). Other constructs including the full-length and various N-terminal truncated human GLP-1R and GCGR were cloned into pcDNA3.1 vector for cAMP accumulation and whole cell-binding assays (Supplementary Table 6).

**Formation and purification of complex**. The Bac-to-Bac Baculovirus Expression System (Invitrogen) was used to generate high-titer recombinant baculovirus for GLP-1R-LgBiT-2MBP, DNGαs, Gβ1-HiBiT, and Gγ2. P0 viral stock was produced by transfecting 5 μg recombinant bacmids into *Sf*9 cells (2.5 mL, density of 1 million cells per mL) for 96 h incubation and then used to produce P1 and P2 baculovirus. GLP-1R-LgBiT-2MBP, DNGαs, Gβ1-HiBiT, and Gγ2 were co-expressed at multiplicity of infection (MOI) ratio of 1:1:1:1 by infecting *Sf*9 cells at a density of 3 million cells per mL with P2 baculovirus (viral titers>90%). The culture was harvested by centrifugation for 48 h post-infection and cell pellets were stored at −80 °C until use.

The cell pellets were thawed and lysed in a buffer containing 20 mM HEPES, pH 7.5, 100 mM NaCl, 10% (v/v) glycerol, 10 mM $MgCl_2$, 1 mM $MnCl_2$, and 100 μM TCEP supplemented with EDTA-free protease inhibitor cocktail (Bimake) by

Dounce homogenization. The complex formation was initiated by the addition of 10 μM GLP-1 or LY3502970 and/or 50 μM compound 2, 10 μg/mL Nb35, and 25 mU/mL apyrase (New England Bio-Labs). After 1.5 h incubation at room temperature (RT), the membrane was solubilized in the buffer above supplemented with 0.5% (w/v) lauryl maltose neopentyl glycol (LMNG, Anatrace) and 0.1% (w/v) cholesterol hemisuccinate (CHS, Anatrace) for 2 h at 4 °C. The supernatant was isolated by centrifugation at 65,000×g for 30 min and incubated with amylose resin (New England Bio-Labs) for 2 h at 4 °C. The resin was then collected by centrifugation at 500×g for 10 min and washed in gravity flow column (Sangon Biotech) with five column volumes of buffer containing 20 mM HEPES (pH 7.5), 100 mM NaCl, 10% (v/v) glycerol, 5 mM MgCl₂, 1 mM MnCl₂, 25 μM TCEP, 0.1% (w/v) LMNG, 0.02% (w/v) CHS, 2 μM GLP-1 or LY3502970 and/or 10 μM compound 2, followed by washing with fifteen column volumes of buffer containing 20 mM HEPES (pH 7.5), 100 mM NaCl, 10% (v/v) glycerol, 5 mM MgCl₂, 1 mM MnCl₂, 25 μM TCEP, 0.03% (w/v) LMNG, 0.01% (w/v) glyco-diosgenin (GDN, Anatrace), 0.008% (w/v) CHS, 2 μM GLP-1 or LY3502970 and/or 10 μM compound 2. The protein was then incubated overnight with TEV protease (customer-made) on the column to remove the C-terminal 2MBP-tag in the buffer above at 4 °C. The flow-through was collected the next day and concentrated with a 100 kDa molecular weight cut-off concentrator (Millipore). The concentrated product was loaded onto a Superdex 200 increase 10/300 GL column (GE Healthcare) with running buffer containing 20 mM HEPES (pH 7.5), 100 mM NaCl, 10 mM MgCl₂, 100 μM TCEP, 1 μM GLP-1 or LY3502970 and/or 5 μM compound 2, 0.00075% LMNG, 0.00025% GDN and 0.0002% (w/v) CHS. The fractions for the monomeric complex were collected and concentrated to 15-20 mg/mL for electron microscopy examination.

**Expression and purification of Nb35.** Nb35 with a C-terminal 6× His-tag was expressed in the periplasm of *E. coli* BL21 (DE3), extracted, and purified by nickel affinity chromatography[52]. Briefly, the Nb35 target gene was transformed in BL21 and grown in a TB culture medium with 100 μg/mL ampicillin, 2 mM MgCl₂, and 0.1% (w/v) glucose at 37 °C, 180 rpm. The expression was induced by adding 1 mM IPTG when OD600 reached 0.7–1.2. The cell pellet was collected by centrifugation after overnight incubation at 28 °C, 180 rpm, and stored at −80 °C until use. The HiLoad 16/600 Superdex 75 column (GE Healthcare) was used to separate the monomeric fractions of Nb35 with running buffer containing 20 mM HEPES, pH 7.5, and 100 mM NaCl. The purified Nb35 was flash-frozen in 30% (v/v) glycerol by liquid nitrogen and stored at −80 °C until use.

**Cryo-EM data acquisition.** The concentrated sample (3.5 μL) was applied to glow-discharged holey carbon grids (Quantifoil R1.2/1.3, 300 mesh), and subsequently vitrified using a Vitrobot Mark IV (ThermoFisher Scientific) set at 100% humidity and 4 °C. Cryo-EM images were collected on a Titan Krios microscope (FEI) equipped with a Gatan energy filter and K3 direct electron detector and performed using serialEM. The microscope was operated at 300 kV accelerating voltage and a calibrated magnification of ×81,000 corresponding to a pixel size of 1.045 Å. The total exposure time was set to 7.2 s with intermediate frames recorded every 0.2 s, resulting in an accumulated dose of 80 electrons per Å² with a defocus range of −1.2 to −2.2 μm. Totally, 3,609 images for compound 2−GLP-1−GLP-1R−Gs, 5,536 images for compound 2−LY3502970−GLP-1R−Gs, and 4,247 images for compound 2−GLP-1−GLP-1R−Gs complexes were collected.

**Image processing.** Dose-fractionated image stacks were subjected to beam-induced motion correction using MotionCor2 v1.4.2[53]. A sum of all frames, filtered according to the exposure dose, was used for further processing in each image stack. Contrast transfer function (CTF) parameters for each micrograph were determined by Gctf v1.06[54]. Particle selection, 2D and 3D classifications were performed on a binned dataset with a pixel size of 2.09 Å using RELION-3.0.8-beta2[55].

For the compound 2–GLP-1–GLP-1R–Gs complex, auto-picking yielded 1,186,340 particle projections were subjected to 3D classification with a mask on the receptor to discard false-positive particles or particles categorized in poorly defined classes, producing 703,620 particle projections for further processing. This subset of particle projections was subjected to further 3D auto-refinement with a mask on the complex, which was subsequently subjected to a round of 3D classifications with a mask on the ECD. A selected subset containing 614,978 projections was subsequently subjected to 3D refinement with a mask on the complex and Bayesian polishing with a pixel size of 1.045 Å. After the last round of refinement, the final map has an indicated global resolution of 2.5 Å at a Fourier shell correlation (FSC) of 0.143. Local resolution was determined using the Bsoft package with half maps as input maps[56].

For the compound 2–GLP-1R–Gs complex, auto-picking yielded 2,366,210 particles. Among them, 21.23% presented better densities on the ECD than other classifications. Thus, this subset was subjected to 3D classification with a mask focused on the ECD and the TMD. Then 285,885 particles were selected for further 3D classification with a mask focused on the ECD. Finally, 147,173 particles were subjected to 3D refinement with a mask focused on the complex and Bayesian polishing with a pixel size of 1.045 Å. The final map has an indicated global resolution of 3.3 Å at an FSC of 0.143. Apart from that, there were 63.42% of the

2,366,210 particles holding better TMD and G protein densities. They were used for 3D classification with masks focused on the TMD and the G protein. Of these, 848,918 particles were selected for 3D classification focused on the TMD. Finally, 340,501 particles were used for final 3D refinement with a mask focused on the complex and Bayesian polishing with a pixel size of 1.045 Å. The final map has an indicated global resolution of 2.5 Å at an FSC of 0.143.

For compound 2–LY3502970–GLP-1R–Gs complex, CTF parameters were estimated with CTFFIND43. A total of 6,566,177 particles were automatically picked from 5,552 images. Particle selection, 2D classification, 3D classification, and refinement were performed using RELION-3.0.8-beta2. A data set of 345,411 particles was subjected to 3D refinement, yielding a final map with a global nominal resolution at 2.9 Å by the 0.143 criteria of the gold-standard FSC. Half-reconstructions were used to determine the local resolution of each map.

**Model building and refinement.** The structures of the LSN3160440–GLP-1–GLP-1R–Gs (PDB: 6VCB), GLP-1–GLP-1R–Gs (PDB: 6X18), and RGT1383–GLP-1R–Gs (PDB: 6B3J) were used as an initial template for model building of compound 2, compound 2 plus GLP-1, and compound 2 plus LY3502970 bound complexes, respectively. Lipid coordinates and geometry restraints were generated using Phenix 1.16. Models were docked into the EM density map using UCSF Chimera 1.13.1. This starting model was then subjected to iterative rounds of manual adjustment and automated refinement in Coot 0.9.4.1 and Phenix 1.16, respectively. The final refinement statistics were validated using the module comprehensive validation (cryo-EM) in Phenix 1.16. Structural figures were prepared in UCSF Chimera 1.13.1 and PyMOL 2.1 (https://pymol.org/2/). The final refinement statistics are provided in Supplementary Table 1.

**cAMP accumulation assay.** Peptide or small molecules stimulated cAMP accumulation was measured by a LANCE Ultra cAMP kit (PerkinElmer). Briefly, after 24 h transfection with various constructs, HEK 293T cells were digested by 0.2% (w/v) EDTA and washed once with Dulbecco's phosphate-buffered saline (DPBS). Cells were then resuspended with stimulation buffer (Hanks' balanced salt solution (HBSS) supplemented with 5 mM HEPES, 0.5 mM IBMX, and 0.1% (w/v) BSA, pH 7.4) to a density of 0.6 million cells per mL and added to 384-well white plates (3,000 cells per well). Different concentrations of ligand in stimulation buffer were added and the stimulation lasted for 40 min at RT. The reaction was stopped by adding a 5 μL Eu-cAMP tracer and ULight-anti-cAMP. After 1 h incubation at RT, the plate was read by an Envision plate reader (PerkinElmer) to measure TR-FRET signals (excitation: 320 nm, emission: 615 nm, and 665 nm). A cAMP standard curve was used to convert the fluorescence resonance energy transfer ratios (665/615 nm) to cAMP concentrations.

**Whole-cell binding assay.** CHO-K1 cells were seeded to 96-well plates (PerkinElmer) coated with fibronectin (Corning) at a density of 30,000 cells per well and incubated overnight. After 24 h transfection, cells were washed twice and incubated with blocking buffer (F12 supplemented with 33 mM HEPES, and 0.1% (w/v) BSA, pH 7.4) for 2 h at 37 °C. Then, radiolabeled ¹²⁵I-GLP-1 (40 pM, PerkinElmer) and increasing concentrations of unlabeled peptide were added and competitively reacted with the cells in binding buffer (PBS supplemented with 10% (w/v) BSA, pH 7.4) at RT for 3 h. After that, cells were washed with ice-cold PBS and lysed by 50 μL lysis buffer (PBS supplemented with 20 mM Tris-HCl and 1% (v/v) Triton X-100, pH 7.4). Finally, 150 μL of scintillation cocktail (OptiPhase SuperMix, PerkinElmer) was employed and radioactivity (counts per minute, CPM) determined by a scintillation counter (MicroBeta2 plate counter, PerkinElmer).

**Molecular dynamics simulation.** Molecular dynamic simulations were performed by Gromacs 2018.5. The receptor was prepared and capped by the Protein Preparation Wizard (Schrodinger 2017-4), while the titratable residues were left in their dominant state at pH 7.0. The complexes were embedded in a bilayer composed of 167 POPC lipids, 42 cholesterols and solvated with 0.15 M NaCl in explicitly TIP3P waters using CHARMM-GUI Membrane Builder v3.2.2[57]. The CHARMM36-CAMP force filed[58] was adopted for protein, peptides, lipids, and salt ions. Compound 2 was modeled with the CHARMM CGenFF small-molecule force field, program version 1.0.0[59]. The Particle Mesh Ewald (PME) method was used to treat all electrostatic interactions beyond a cutoff of 10 Å and the bonds involving hydrogen atoms were constrained using the LINCS algorithm[60]. The complex system was firstly relaxed using the steepest descent energy minimization, followed by slow heating of the system to 310 K with restraints. The restraints that adopted from the default setting in the CHARM-GUI webserver v3.2.2[57] were reduced gradually over 50 ns, with a simulation step of 1 fs (see Supplementary Table 5 for more details). Finally, a restraint-free production run was carried out for each simulation, with a time step of 2 fs in the NPT ensemble at 310 K and 1 bar using the Nose-Hoover thermostat and the semi-isotropic Parrinello-Rahman barostat[61], respectively. The interface areas of agonist-TMD and ECD-TMD were calculated using FreeSASA 2.0[62] with the Sharke-Rupley algorithm in a probe radius of 1.2 Å.

**Statistical analysis.** All functional study data were analyzed using Prism 8.0 (GraphPad) and presented as means ± S.E.M. from at least three independent

experiments. Concentration-response curves were evaluated with a three-parameter logistic equation. The significance was determined with either a two-tailed Student's $t$-test or one-way ANOVA, and $P < 0.05$ was considered statistically significant.

**Reporting summary.** Further information on research design is available in the Nature Research Reporting Summary linked to this article.

## Data availability

All relevant data are available from the corresponding authors upon reasonable request. The raw data underlying Figs. 2c, 3b, and Supplementary Figs. 1b–e, 5c, 8a are provided as a Source Data file. The atomic coordinates and electron microscopy maps have been deposited in the Protein Data Bank (PDB) under accession codes: 7DUR (compound 2–GLP-1R–G$_s$ complex with ECD), 7EVM (compound 2–GLP-1R–G$_s$ complex without ECD), 7DUQ (compound 2–GLP-1–GLP-1R–G$_s$ complex) and 7E14 (compound 2–LY3502970–GLP-1R–G$_s$ complex) and Electron Microscopy Data Bank (EMDB) accession codes: EMD-30867 (compound 2–GLP-1R–G$_s$ complex with ECD), EMD-31329 (compound 2–GLP-1R–G$_s$ complex without ECD), EMD-30886 (compound 2–GLP-1–GLP-1R–G$_s$ complex) and EMD-30936 (compound 2–LY3502970–GLP-1R–G$_s$ complex), respectively. Source data are provided with this paper.

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

## Acknowledgements

We thank D.P. Yuan, W. Guo, F. Liu, B. Qiu, P.F. Lan, and M. Lei for technical advice. This work was partially supported by National Natural Science Foundation of China 81872915 (M.-W.W.), 82073904 (M.-W.W.), 81922071 (Y.Z.), 81773792 (D.Y.), 81973373 (D.Y.), and 21704064 (Q.Z.); National Science and Technology Major Project of China–Key New Drug Creation and Manufacturing Program 2018ZX09735–001 (M.-W.W.) and 2018ZX09711002–002–005 (D.Y.); the National Key Basic Research Program of China 2018YFA0507000 (M.-W.W.) and 2019YFA0508800 (Y.Z.), Shanghai Municipal Science and Technology Major Project 2019SHZDZX02 (H.E.X.); Ministry of Science and Technology of China Major Project XDB08020303 (H.E.X.); Shanghai Municipal Science and Technology Commission grant 19ZR1467500 (H.L.M.); Zhejiang Province Science Fund for Distinguished Young Scholars LR19H310001 (Y.Z.); Fundamental Research Funds for Central Universities 2019XZZX001-01-06 (Y.Z.); Novo Nordisk-CAS Research Fund grant NNCAS-2017–1-CC (D.Y.); The Young Innovator Association of CAS Enrollment (H.L.M. and L.H.Z.) and SA-SIBS Scholarship Program (L.H.Z. and D.Y.). The cryo-EM data were collected at Cryo-Electron Microscopy Research Center, Shanghai Institute of Materia Medica, Chinese Academy of Sciences.

## Author contributions

Z.T.C. and H.L.M. designed the expression constructs, purified the receptor complexes, screened specimen, prepared the final samples for negative staining/data collection towards the structure, and participated in manuscript preparation; L.-N.C., H.L.M., and X.Y.Z. performed map calculation, structure analysis, figure preparation and participated in manuscript writing; Q.T.Z. performed structural analysis, MD simulations, figure preparation and participated in manuscript writing; Q.L. synthesized compound 2; Z.T.C. and A.T.D. conducted ligand binding and signaling experiments; C.Y.Y. and X.W. assisted in complex purification; T.X. assisted in structural analysis; L.H.Z. and P.Y.X. helped construct design and data analysis; W.H., X.Q.S., and W.G.Z. supplied LY3502970 and helped LY3502970 related structure determination; D.Y. supervised mutagenesis and signaling experiments, participated in data analysis and manuscript preparation; H.E.X., Y.Z., W.G.Z., and M.-W.W. initiated the project, supervised the studies, analyzed the data, and wrote the manuscript with inputs from all co-authors.

## Competing interests

The authors declare no competing interests.
