## [Peer Review File · Nature Communications]

REVIEWER COMMENTS

Reviewer #1 (Remarks to the Author):

In their important and timely manuscript: "Molecular insights into ago-allosteric modulation of the human glucagon-like peptide-1 receptor" Cong et al. investigate the allosteric regulation of glucagon-like peptide-1 (GLP-1) receptor by resolving 3 high quality cryo-EM structures of GLP-1R bound to an allosteric compound alone or in combination with GLP-1 or a small molecule agonist (OWL833). All the complexes are bound to heterotrimeric G proteins.

GLP-1R is not only a key regulator of glucose homeostasis and as such a validated drug target for type 2 diabetes and obesity, but it is also a paradigmatic representative of class B1 G-protein-coupled receptors. Thus, understanding its allosteric regulation by ago-allosteric modulators (as the one investigated here) is of great importance both from a fundamental and translational point of view.

The high resolution cryo-EM structures and the complementary MD simulations performed by the authors reveal an interesting mechanism whereby the allosteric modulator covalently bound to C347 triggers the outward movement of TM6 and a downward shift of the N-terminal α -helix of the ECD explaining the observed positive allosteric modulation on both agonist binding and G protein coupling. This kind of detailed structural and mechanistic insight of ago-allosterism in the GLP-1R might allow the rational design of novel therapeutic agents.

The main conclusions of the papers are supported by the data. However, there is a related question that is part of the activation mechanism which is only tangentially touched by the paper (in particular Fig. 6 and discussion on pag. 10) but should be better clarified in both the figure and the text to avoid misinterpretation of the data.

The question is the role of the G protein in the full activation of the GLP-1R. Recent papers using FRET, simulations and other techniques (see e.g. Hilger et al. Science 369, 2020 DOI: 10.1126/science.aba3373 and Mattedi et al. PNAS 117, 2020 - Ref. 40) show that the G-protein can have a direct role in inducing the full activation of the receptor. This of course does not exclude a partial opening of the intracellular side in the sole presence of the GLP and the allosteric modulator, but the population of this open state in other receptors (including the Glucagon receptor) has been shown to be low and various lines of evidence concur to show that the full activation is 'induced' by the binding of the Gs. As the cryo-EM structures reported here are all in complex with Gs, the authors cannot probably directly comment on this issue. However, they could cite the recent literature and clarify this point in the text. In particular the text on pag. 10 and the caption of the figure 6 (currently "an open conformation, which in turn creates an intercellular binding site for G protein coupling") should reflect the recent findings and be revised to reflect them (e.g. "which in turn lowers the energy barrier"- or creates the conditions for the binding...).

Reviewer #2 (Remarks to the Author):

The article by Z. Cong et al is the first report of a high-resolution structure of an ago-allosteric modulator covalently bound to the GLP-1R. This paper is well-written, and the data come from several leading scientists in this area. The experiments are well-performed, and the findings appear to be of high quality. Overall, this article has the potential to be a significant contribution, but there are some aspects of the paper that could be improved.

Major Points:

1. The authors should include a Figure in the paper indicating the density fit of compound 2, the GLP-1 peptide, and OWL833.
2. The structure with compound 2 alone is of much lower resolution ($\sim 3.4\text{\AA}$) than the other two

structures. Upon examining the density, one cannot really see the density of compound 2 for the compound 2-only structure. Since all three structures are prepared with compound 2 and the compound 2-only structure is in the active conformation, it can be assumed that compound 2 is present in the compound 2-only structure, but an alternate verification is needed (since clear density for the ligand is not observed). One possibility is to provide mass spectrometry verification that the receptor is indeed modified by compound 2 at Cys347.

3. Similarly, although the cryoEM structures of the GLP-1R bound to compound 2 in the current manuscript indicate C347 forms a covalent adduct with the compound, this covalent mechanism of action using C347 was previously shown for BETP (another reactive PAM) using LC mass spectrometry (PMID: 24997604). The pharmacology/mutagenesis studies in this previous report supported the same covalent mechanism for compound 2 (i.e. via C347). In this previous paper, the mass spectrometry studies also indicated that C438 formed a covalent adduct with BETP. In the cryoEM studies in this report, C438 is disordered. Therefore, the authors of the current paper should verify whether modification of C438 occurs. The authors should also more thoroughly discuss their findings in context of the previous report that made this original discovery.

4. Previous studies reported the structure-activity relationship (SAR) for compound 2 (PMID: 17827014). Therefore, to better validate the cryoEM findings that are presented in the manuscript, the authors should provide an analysis of the proposed binding mechanism of compound 2 in the context of the SAR. The authors point out in their abstract that the data in their report could enable small molecule drug discovery and the SAR analyses could help inspire new work in this space.

5. Both the compound 2-only structure and the OWL833 + compound 2 structure share a similar contact between the ECD and ECL1. The contact is hydrophobic and is composed of W39 (ECD), Y88 (ECD), and W214 (ECL1). Interestingly, in the structure containing the GLP-1 peptide, the contact with W214 (ECL1) is replaced by F28 and W31 of the peptide. Data in the manuscript show that truncation of the ECD reduces compound 2 activity. To better refine the hypothesis that interaction of the ECD with ECL1 is important for receptor activation, cAMP experiments with single mutants of W39A, Y88A, and W214A in the GLP-1R should be performed. These functional data would support the overall structural findings.

6. In the cryoEM structures, several cholesterol molecules are observed. Are these from the GDN in the buffer? In the other similarly high resolution structures reported in the literature (GLP-1R-GLP-1 and GLP-1R-CHU-128; PMID 33027691), no cholesterol molecules were modeled, and those investigators did not use GDN in the buffer. Could the cholesterol molecules in your structures be an artifact that arose from the GDN in the buffer? How can it be certain that the cholesterol molecules around TM5 and TM6 are stabilized by the presence of compound 2? This is an important issue that should be discussed and clarified in the text.

7. In line 438, please indicate the rationale for including 42 cholesterol in MD? Also, how were the cholesterol placed in the receptor complex initially?

8. The authors should provide RSCC (real space correlation coefficient) calculations for compound 2 and OWL833 in all cryoEM structure models.

Minor Points:

1. To avoid confusion in the field, the nomenclature of the compound "OWL833" should be changed to "LY3502970". This compound is currently in clinical development under LY3502970, and several trials are listed under this name in clinicaltrials.gov. Since the compound was known as OWL833 prior to entering clinical development, this should be acknowledged by saying "LY3502970 (OWL833)" at the first mention of the compound in the paper (i.e. in the abstract). This approach was taken in the initial publication of the compound (PMID: 33177239). Also, it should be noted that a high-resolution

structure of LY3502970 (OWL833) in complex with the GLP-1R was reported in this paper (last November). The authors should more clearly reference this cryoEM structure/article in the current manuscript.

2. In lines 389 and 395, the structure with compound 2-only was processed with two sets of data, with the final map processed to 2.8Å. However, the uploaded mrc map file appears to be at 3.4Å (not at 2.8Å). Please double check the resolution of the uploaded map for the structure with compound 2-only.

3. In lines 169, 171, 176, and 177, the authors discuss truncating the ECD at residues 28, 33, 48, and 55. Is this counting the signal peptide in the residue count? Please provide the exact amino acid boundary of the N-terminal truncations in Figure 3.

4. In line 445, please indicate the level and on which atoms of the restraints were applied and changed.

5. In Supplementary Figure 7. "all snapshots obtained from MD simulations were superimposed on Ca atoms of the compound 2-GLP-1R-Gs complex". Were all Ca atoms, including both ECD and TMD, used for the protein superimposition?

6. In Supplementary Figures 7 and 9. Figure 7a compares MD snapshots with the cryoEM structure. Is the reference structure for calculating Figures 7b Ca RMSD plots also the cryoEM structure or the frame 0? If not the cryoEM structure, please indicate the RMSD between the last MD snapshot and the cryoEM structure.

7. In Supplementary Figure 9b. Please explain why the Ca RMSD of the ECD gradually reduces over MD.

Kyle Sloop

Reviewer #3 (Remarks to the Author):

Zhaotong Cong and colleagues present in their manuscript three GLP-1R complex cryo-EM structures in the presence of the ago-allosteric modulator "compound 2". Their results provide new structural insights into the agonist and allosteric action of this compound and thereby contribute valuable information towards possible pharmacological applications of such agents. Of special notice is the proposed cooperation of the ECD as an agonist in the absence of orthosteric ligand. The paper will be of great interest among the structural, GPCR, and drug development communities and, in principle, I would recommend it for publication, if the authors can address the following issues:

1. The cryo-EM densities for compound 2 in Suppl. Fig. 3 look quite ambiguous. To provide visual evidence for the statement on line 129 "cryo-EM map demonstrates that compound 2 is covalently bound to C347", please add a new supplementary figure showing zoomed-in views of the compound and the interacting region of TM6 placed inside the density maps of the three samples.
2. Line 141, Fig. 2c: A350W and especially K351A seem to not just reduce the response but diminish it almost as much as the C347A mutation. Please rephrase the sentence to reflect the strong effect of these mutations.
3. The presence of another small GLP-1R agonist molecule (WB4-24) in the compound 2-GLP-1R sample is mentioned just in the cryo-EM data acquisition section. At which step was WB4-24 added and in what concentration? It was not mentioned in the complex formation and purification part. Is the binding site of WB4-24 known? If yes, please include a new panel in Suppl. Fig. 11 showing the WB4-24 model imposed on a zoomed-in view of both the global and ECD-resolved maps to reaffirm the

absence of density at that location in both maps. Is it possible that WB4-24 contributes to the stabilization of the observed ECD conformation with N-term. helix insertion into the TMD core?

4. Which map is shown in Fig. 1a? It does not look like either of the maps in Suppl. Fig. 2a.

5. Was the compound 2 density present in both the ECD-focused and the global maps? Please deposit both maps in EMDB. Currently, there is only one accession number listed in the data availability section.

6. There are no dataset and processing details for the additional data that was collected for the compound 2-GLP-1R and according to the cryo-EM data acquisition section produced a 3.4 Å map. This map was used to confirm similarity with the WB4-24-containing sample map. If the data and map were used in the work, I would strongly encourage including the experimental and processing details in the paper and depositing the additional map in EMDB.

7. It is surprising that the particles used for the ECD-focused map and the particles used for the global map were separate subsets (Suppl. Fig. 2a). Typically, the active state will have the most rigid and well-resolved structure and according to the presented hypothesis, the ECD insertion into the TMD should have further stabilized the global structure. Did the authors try combining the two subsets together to possibly get a better global map and then classifying-out a subset with an ECD-focused mask?

8. Was density for the ECD N-term. helix tip entering the TMD core also present in the current global map (Suppl. Fig. 2a, right)? If not, is the inserted ECD only a transient state that further stimulates G protein association? Please provide a comment on this in the discussion.

Minor corrections:

- The FSC curves in Suppl. Fig. 2c seem to be misrepresented, FSCwork and FSCtest in particular. Were the Relion FSC curves for uncorrected/noise-substituted maps misplaced as FSCwork and FSCtest?

- Line 62: ref.19 does not summarize GLP-1R peptide and small molecule agonist structural studies, as implied in the sentence.

- Suppl. Fig. 2: please add class distribution values to the workflow graphs

- Suppl. Fig. 3: for easier interpretation and to avoid confusion, I would suggest presenting the structures in the same order as in Fig. 1 and Suppl. Fig. 2.

- Line 143: "evaluated" -> "elevated"

- Line 229: "their overall structure" -> "the overall structure"

- Suppl. Material, line 52: Compassion -> Comparison

- Suppl. Table 1: The electron exposures are listed as 25.3 e/Å² while in the methods section it is 80 e/Å².

POINT-BY-POINT RESPONSES TO THE REVIEWERS' COMMENTS

Reviewer #1 (Remarks to the Author):

In their important and timely manuscript: “Molecular insights into ago-allosteric modulation of the human glucagon-like peptide-1 receptor” Cong et al. investigate the allosteric regulation of glucagon-like peptide-1 (GLP-1) receptor by resolving 3 high quality cryo-EM structures of GLP-1R bound to an allosteric compound alone or in combination with GLP-1 or a small molecule agonist (OWL833). All the complexes are bound to heterotrimeric G proteins.

GLP-1R is not only a key regulator of glucose homeostasis and as such a validated drug target for type 2 diabetes and obesity, but it is also a paradigmatic representative of class B1 G-protein-coupled receptors. Thus, understanding its allosteric regulation by ago-allosteric modulators (as the one investigated here) is of great importance both from a fundamental and translational point of view.

The high resolution cryo-EM structures and the complementary MD simulations performed by the authors reveal an interesting mechanism whereby the allosteric modulator covalently bound to C347 triggers the outward movement of TM6 and a downward shift of the N-terminal α -helix of the ECD explaining the observed positive allosteric modulation on both agonist binding and G protein coupling. This kind of detailed structural and mechanistic insight of ago-allosterism in the GLP-1R might allow the rational design of novel therapeutic agents.

The main conclusions of the papers are supported by the data. However, there is a related question that is part of the activation mechanism which is only tangentially touched by the paper (in particular Fig. 6 and discussion on page 10) but should be better clarified in both the figure and the text to avoid misinterpretation of the data.

The question is the role of the G protein in the full activation of the GLP-1R. Recent papers using FRET, simulations and other techniques (see e.g. Hilger et al. Science 369, 2020 DOI: 10.1126/science.aba3373 and Mattedi et al. PNAS 117, 2020 - Ref. 40) show that the G-protein can have a direct role in inducing the full activation of the receptor. This of course does not exclude a partial opening of the intracellular side in the sole presence of the GLP and the allosteric modulator, but the population of this open state in other receptors (including the glucagon receptor) has been shown to be low and various lines of evidence concur to show that the full activation is ‘induced’ by the binding of the Gs. As the cryo-EM structures reported here are all in complex with Gs, the authors cannot probably directly comment on this issue. However, they could cite the recent literature and clarify this point in the text. In particular the text on page 10 and the caption of the figure 6

Response: We totally agreed with this reviewer’s valuable suggestion. The recent publications have been included in our revised manuscript. Besides, the “G_s” label has been added in Figure 6d-f to show the importance of G protein in compound 2-induced receptor activation. Meanwhile, the main text and figure legends have been updated to reflect these points.

(Currently “an open conformation, which in turn creates an intercellular binding site for G protein coupling”) should reflect the recent findings and be revised to reflect them (e.g. “which

in turn lowers the energy barrier”- or creates the conditions for the binding...).

Response: We thank the reviewer for this comment which is well-taken and relevant revision has been made in the revised text

Reviewer #2 (Remarks to the Author):

The article by Z. Cong et al is the first report of a high-resolution structure of an ago-allosteric modulator covalently bound to the GLP-1R. This paper is well-written, and the data come from several leading scientists in this area. The experiments are well-performed, and the findings appear to be of high quality. Overall, this article has the potential to be a significant contribution, but there are some aspects of the paper that could be improved.

Major Points:

1. The authors should include a Figure in the paper indicating the density fit of compound 2, the GLP-1 peptide, and OWL833.

Response: This point is well taken, and the density fit of the ligands (as shown below) were provided in the revised Supplementary Fig. 3.

Supplementary Fig. 3 Near-atomic resolution model of the complexes in the cryo-EM density maps. a, EM density map and model of the compound 2–GLP-1R– G_s complex are shown for all seven-transmembrane (7TM) α -helices, helix 8 and all extracellular loops of GLP-1R, compound 2 and the α_5 -helix of the $G\alpha_s$ Ras-like domain. **b**, EM density map and model of the compound 2–GLP-1–GLP-1R– G_s complex are shown for all 7TM α -helices, helix 8 and all extracellular loops of GLP-1R, GLP-1, compound 2 and the α_5 -helix of the $G\alpha_s$ Ras-like domain. **c**, EM density map and model of the compound 2–LY3502970–GLP-1R– G_s complex are shown for all 7TM α -helices, helix 8 and ECL1 of GLP-1R, compound 2, LY3502970 and the α_5 -helix of the $G\alpha_s$ Ras-like domain. **d**, The zoomed-in views of compound 2 and the interacting region of TM6 placed inside the density maps of compound 2–GLP-1 (left), compound 2–GLP-1–GLP-1R (middle) and compound 2–LY3502970–GLP-1R (right).

2. The structure with compound 2 alone is of much lower resolution ($\sim 3.4\text{\AA}$) than the other two structures. Upon examining the density, one cannot really see the density of compound 2 for the compound 2-only structure. Since all three structures are prepared with compound 2 and the

compound 2-only structure is in the active conformation, it can be assumed that compound 2 is present in the compound 2-only structure, but an alternate verification is needed (since clear density for the ligand is not observed). One possibility is to provide mass spectrometry verification that the receptor is indeed modified by compound 2 at Cys347.

Response: We thank the reviewer for the comments. Actually, we have prepared the compound 2 alone sample again and resolved the complex structure showing a clear density of compound 2 in the map (see revised Supplementary Fig. 3 above). Thus, we replaced the previous 3.4Å structure with this newly-solved cryo-EM structure of compound 2-bound GLP-1R (2.5Å).

3. Similarly, although the cryoEM structures of the GLP-1R bound to compound 2 in the current manuscript indicate C347 forms a covalent adduct with the compound, this covalent mechanism of action using C347 was previously shown for BETP (another reactive PAM) using LC mass spectrometry (PMID: 24997604). The pharmacology/mutagenesis studies in this previous report supported the same covalent mechanism for compound 2 (i.e. via C347). In this previous paper, the mass spectrometry studies also indicated that C438 formed a covalent adduct with BETP. In the cryoEM studies in this report, C438 is disordered. Therefore, the authors of the current paper should verify whether modification of C438 occurs. The authors should also more thoroughly discuss their findings in context of the previous report that made this original discovery.

Response: We appreciate very much this reviewer's insightful comments. As the reviewer suggested, we performed the LC mass spectrometry experiments. Since it is a high precision quantitative method that we are not familiar with, several technological hurdles prevented us from solving the issue in a timely manner.

As far as C347 is concerned, our newly obtained data clearly demonstrate a continuous EM density between compound 2 and C347, even with an increasing contour threshold, allowing us to build an accurate model showing covalent modification of compound 2 (see figure below). This is consistent with two previous studies (PMID: 24997604 and PMID: 26975372) involving BETP and its analogues including compound 2.

Continuous EM density of compound 2 in the compound 2-GLP-1-GLP-1R complex structure. The receptor is shown in red and compound 2 in orange. G protein is omitted for clarity.

In the case of C438, we carried out single-point mutation experiment to investigate the effect of C438A mutant on compound 2-induced cAMP responses. In line with the previous report showing that mutation of C438 did not alter the PAM activity of compound 2 (PMID:

24997604), we found that cAMP accumulation elicited by C438A was similar to that of the wild-type (WT, see figure below). However, we cannot rule out possible formation of a covalent adduct at C348. Thus, we discussed this concern in the revised text: “The same covalent mechanism of action was previously demonstrated for another PAM of GLP-1R (i.e., BETP), showing that in addition to C347, BETP also formed a covalent adduct with C438 in the C terminus²⁵ that is invisible in the current GLP-1R complex structures. Although mutation of C438 did not alter the PAM activity of BETP or compound 2 (Ref. ²⁵), it cannot rule out possible formation of a covalent adduct at C348 in our compound 2-bound GLP-1R structure.”

Effect of C438A mutation on compound 2-induced cAMP accumulation.

4. Previous studies reported the structure-activity relationship (SAR) for compound 2 (PMID: 17827014). Therefore, to better validate the cryo-EM findings that are presented in the manuscript, the authors should provide an analysis of the proposed binding mechanism of compound 2 in the context of the SAR. The authors point out in their abstract that the data in their report could enable small molecule drug discovery and the SAR analyses could help inspire new work in this space.

Response: We appreciate these comments and have revised our manuscript accordingly. “Instead of occupying the orthosteric binding pocket where peptide or small molecule agonists generally bind, the high-resolution cryo-EM map demonstrates that compound 2 is covalently bonded to C347^{6.36b} (Wootten numbering in superscript³⁶) and mounted on the membrane-facing surface of the cytoplasmic end of TM6, providing solid structural evidence of a unique binding site for the ago-allosteric modulator (Fig. 2b). Such a covalent modification is supported by a previous report showing that non-sulfonic substituents at C-2 position (methylsulfone) failed to produce measurable cAMP responses³⁶, consistent with our mutagenesis results, where C347^{6.36b}A mutation diminished the potency of compound 2 without affecting that of GLP-1 (Fig. 2c and Supplementary Table 2). Compound 2 forms predominantly hydrophobic interactions with the adjacent residues in TM6. The tert-butyl moiety of compound 2 points to TM7 and makes hydrophobic contacts with A350^{6.39b} and K351^{6.40b}, replacement at C-3 position by polar functional groups caused dramatic decline in its potency and efficacy³⁶. The dichloroquinoxaline group extends to ICL3 forming nonpolar interactions with K346^{6.35b}, C347^{6.36b} and a cholesterol molecule in TM6, introduction of electron-donating substituents or replacement of quinoxalines by benzimidazole or quinoline at C-6 and C-7 positions led to poor tolerance³⁶.”

5. Both the compound 2-only structure and the OWL833 + compound 2 structure share a similar

contact between the ECD and ECL1. The contact is hydrophobic and is composed of W39 (ECD), Y88 (ECD), and W214 (ECL1). Interestingly, in the structure containing the GLP-1 peptide, the contact with W214 (ECL1) is replaced by F28 and W31 of the peptide. Data in the manuscript show that truncation of the ECD reduces compound 2 activity. To better refine the hypothesis that interaction of the ECD with ECL1 is important for receptor activation, cAMP experiments with single mutants of W39A, Y88A, and W214A in the GLP-1R should be performed. These functional data would support the overall structural findings.

Response: To answer this question, we performed additional experiments to reveal the functional role of the ECD-ECL1 interaction in the presence of compound 2. The results demonstrate that all the three mutants reduced cAMP potency (see figure below), indicating that this interaction is important for receptor activation.

Supplementary Fig. 5c Effects of W39A, Y88A, and W214A on compound 2-induced cAMP responses. Data shown are means \pm S.E.M. of three independent experiments. WT, wild-type.

Supplementary Table. 4 Effects of residue mutation in the ECD-ECL1 interface on cAMP accumulation.

	Compound 2	
	pEC_{50}	E_{max}
WT GLP-1R	6.72 ± 0.05	99.99 ± 1.93
W39A	$6.26 \pm 0.12^{**}$	95.79 ± 4.51
Y88A	$5.72 \pm 0.09^{***}$	$84.74 \pm 4.06^*$
W214A	$6.24 \pm 0.11^{**}$	95.54 ± 3.96

cAMP accumulation data were analyzed using a three-parameter logistic equation to determine pEC_{50} and E_{max} values. pEC_{50} is the negative logarithm of the molar concentration of agonist that induced half the maximal response. E_{max} for mutants is expressed as a percentage of the wild-type (WT). All values are means \pm S.E.M. of at least three independent experiments conducted in duplicate. One-way ANOVA was used to determine statistical significance (** $P < 0.01$, * $P < 0.05$).

6. In the cryo-EM structures, several cholesterol molecules are observed. Are these from the GDN in the buffer? In the other similarly high resolution structures reported in the literature (GLP-1R-GLP-1 and GLP-1R-CHU-128; PMID 33027691), no cholesterol molecules were

modeled, and those investigators did not use GDN in the buffer. Could the cholesterol molecules in your structures be an artifact that arose from the GDN in the buffer? How can it be certain that the cholesterol molecules around TM5 and TM6 are stabilized by the presence of compound 2? This is an important issue that should be discussed and clarified in the text.

Response: We thank the reviewer for the comments. Cholesterol has been observed in the cryo-EM structures of several class B1 GPCRs including PTH1R (PDB code: 6NBF), GHRHR (PDB code: 7CZ5), CRF1R (PDB code: 6PB0) and CRF2R (PDB code: 6PB1), as well as some class A GPCRs (e.g., CB1, PDB code: 5XRA and 5XR8; CXCR2, PDB code: 6LFL; 5-HT1A, PDB code: 7E2X and 7E2Z; GABA_B, PDB code: 7C7S and 7C7Q; D1DR, PDB code: 7LJC and 7LJD). Among them, cholesterol molecules were also observed in the cryo-EM structures of CB1, CXCR2, GABA_B and D1DR where no GDN was used in the sample preparation. It thus appears that the presence of cholesterol in some of the structures is not related to the use of GDN.

In our research, we found that there are solid EM densities located in the TM5-TM6 cleft that is close to compound 2 in all three structures (compound 2–GLP-1R–G_s, compound 2–GLP-1–GLP-1R–G_s and compound 2–OWL833–GLP-1R–G_s, see figure below), allowing an accurate modelling of two cholesterol. Given the fact that cholesterol molecules in such specific positions have been only observed in these compound 2-bound structures, we thought it necessary to mention this observation on our manuscript. Since this phenomenon was also noted in our newly solved cryo-EM structure of peptide 20-bound GLP-1R-G_s complex, we performed MD simulations and found that these two cholesterol did not stably sit in the original positions. Therefore, the significance of their presence has not been determined.

Cholesterol binding in the cryo-EM structure of compound 2–GLP-1–GLP-1R complex.

The cryo-EM density map of two cholesterol close to compound 2 is shown in mesh at a 0.01 threshold. The receptor is shown in red, compound 2 in orange, and two cholesterol close to compound 2 in yellow. Receptor ECD and G protein are omitted for clarity.

7. In line 438, please indicate the rationale for including 42 cholesterol in MD? Also, how were the cholesterol placed in the receptor complex initially?

Response: We thank the reviewer for the comments. Given that cholesterol is one important

component in cell membrane (up to 30%) and may involve in the compound 2-bound GLP-1R activation (according to the cryo-EM density), we built a bilayer membrane composed of POPC and cholesterol (mole ratio is 4:1) using the “Membrane Builder” tool in the CHARMM-GUI webserver, where cholesterol is randomly distributed (see “Automated Builder and Database of Protein/Membrane Complexes for Molecular Dynamics Simulations”, *PLoS One*, 2007, 2(9):e880 for details). During our 1 μ s MD simulations, no cholesterol was able to occupy the similar positions of two cholesterol (located in the TM5-TM6 cleft) observed in our cryo-EM structures, while some cholesterol occasionally passed by this region.

8. The authors should provide RSCC (real space correlation coefficient) calculations for compound 2 and OWL833 in all cryo-EM structure models.

Response: We thank the reviewer for the suggestion and hence, calculated the RSCC values for compound 2 (0.59, 0.54, 0.47 and 0.46 in compound 2-GLP-1R without ECD, compound 2-GLP-1R with ECD, GLP-1-compound 2-GLP-1R and OWL833-compound 2-GLP-1R structures, respectively) and OWL833 (0.74 in OWL833-compound2-GLP-1R structure). The results were added to Supplementary Table 1.

Minor Points:

1. To avoid confusion in the field, the nomenclature of the compound “OWL833” should be changed to “LY3502970”. This compound is currently in clinical development under LY3502970, and several trials are listed under this name in clinicaltrials.gov. Since the compound was known as OWL833 prior to entering clinical development, this should be acknowledged by saying “LY3502970 (OWL833)” at the first mention of the compound in the paper (i.e. in the abstract). This approach was taken in the initial publication of the compound (PMID: 33177239). Also, it should be noted that a high-resolution structure of LY3502970 (OWL833) in complex with the GLP-1R was reported in this paper (last November). The authors should more clearly reference this cryo-EM structure/article in the current manuscript.

Response: We thank the reviewer for the suggestion and changed “OWL833” to “LY3502970” globally in our revised manuscript.

2. In lines 389 and 395, the structure with compound 2-only was processed with two sets of data, with the final map processed to 2.8Å. However, the uploaded mrc map file appears to be at 3.4Å (not at 2.8Å). Please double check the resolution of the uploaded map for the structure with compound 2-only.

Response: We thank the reviewer for the comment. The 3.4Å map was used in the structural analysis due to relatively better resolution of the ECD, and thus we uploaded this mrc map in the main file and the 2.8Å map as a supplementary file. To avoid confusion, we replaced both maps with our newly-solved cryo-EM structure of compound 2-bound GLP-1R that holds better density of compound 2 and ECD.

3. In lines 169, 171, 176, and 177, the authors discuss truncating the ECD at residues 28, 33,

48, and 55. Is this counting the signal peptide in the residue count? Please provide the exact amino acid boundary of the N-terminal truncations in Figure 3.

Response: We thank the reviewer for the suggestion. The signal peptide was counted, and the schematic diagram of residue truncations (as shown below) was provided in revised Figure 3b.

Schematic diagram of residue truncations. Δ , residue truncation.

4. In line 445, please indicate the level and on which atoms of the restraints were applied and changed.

Response: We thank the reviewer for the comment. For the restrains, we chose the default setting suggested by the “Membrane Builder” tool in the CHARMM-GUI webserver, which is widely adopted. In detail, the harmonic position restraints of 40 and 20 $\text{kJ}\cdot\text{mol}^{-1}\cdot\text{\AA}^{-2}$ were applied to the backbone and side-chain non-hydrogen atoms of protein and peptide. For POPC lipid, a planar harmonic restraint ($10 \text{kJ}\cdot\text{mol}^{-1}\cdot\text{\AA}^{-2}$) on the phosphorus atom was applied to hold the position of lipid head groups of membranes along the Z-axis, while dihedral restraint ($1,000 \text{kJ}\cdot\text{mol}^{-1}\cdot\text{rad}^{-2}$) was applied on two dihedrals (C28-C29-C210-C211 and C1-C3-C2-O21) to keep fatty acid chain double bonds in the cis conformation and C2 chirality for each lipid molecule. These restraints were gradually reduced to zero and then restrain-free production simulations were performed. The detailed information are provided as below. We revised the “Molecular dynamics simulation” section in the Methods and added the following table as Supplementary Table 5 in the revision.

Supplementary Table 5. Details of restraints applied during MD simulations.

Stage	Time step	Simulation time	Restrain
Heating	1 fs	1 ns	Position harmonic restraint ($40 \text{kJ}\cdot\text{mol}^{-1}\cdot\text{\AA}^{-2}$) for the backbone non-hydrogen atoms of protein and peptide; Position restraint ($20 \text{kJ}\cdot\text{mol}^{-1}\cdot\text{\AA}^{-2}$) for the sidechain non-hydrogen atoms of protein and peptide; Planar harmonic restraint ($10 \text{kJ}\cdot\text{mol}^{-1}\cdot\text{\AA}^{-2}$) for the phosphorus atom of POPC along the Z-axis; Dihedral restraint ($1000 \text{kJ}\cdot\text{mol}^{-1}\cdot\text{rad}^{-2}$) for two dihedrals (C28-C29-C210-C211 and C1-C3-C2-O21).
Step6.1	1 fs	5 ns	Position harmonic restraint ($40 \text{kJ}\cdot\text{mol}^{-1}\cdot\text{\AA}^{-2}$) for the backbone non-hydrogen atoms of protein and peptide; Position restraint ($20 \text{kJ}\cdot\text{mol}^{-1}\cdot\text{\AA}^{-2}$) for the sidechain non-hydrogen atoms of protein and peptide; Planar harmonic restraint ($10 \text{kJ}\cdot\text{mol}^{-1}\cdot\text{\AA}^{-2}$) for the phosphorus atom of POPC along the Z-axis; Dihedral restraint ($1000 \text{kJ}\cdot\text{mol}^{-1}\cdot\text{rad}^{-2}$) for two dihedrals (C28-C29-C210-C211 and C1-C3-C2-O21).
Step6.2	1 fs	5 ns	Position harmonic restraint ($20 \text{kJ}\cdot\text{mol}^{-1}\cdot\text{\AA}^{-2}$) for the backbone non-

			hydrogen atoms of protein and peptide; Position restrain (10 kJ·mol⁻¹·Å⁻²) for the sidechain non-hydrogen atoms of protein and peptide; Planar harmonic restraint (4 kJ·mol⁻¹·Å⁻²) for the phosphorus atom of POPC along the Z-axis; Dihedral restraint (400 kJ·mol⁻¹·rad⁻²) for two dihedrals (C28-C29-C210-C211 and C1-C3-C2-O21).
Step6.3	1 fs	10 ns	Position harmonic restrain (10 kJ·mol⁻¹·Å⁻²) for the backbone non-hydrogen atoms of protein and peptide; Position restrain (5 kJ·mol⁻¹·Å⁻²) for the sidechain non-hydrogen atoms of protein and peptide; Planar harmonic restraint (4 kJ·mol⁻¹·Å⁻²) for the phosphorus atom of POPC along the Z-axis; Dihedral restraint (200 kJ·mol⁻¹·rad⁻²) for two dihedrals (C28-C29-C210-C211 and C1-C3-C2-O21).
Step6.4	1 fs	10 ns	Position harmonic restrain (5 kJ·mol⁻¹·Å⁻²) for the backbone non-hydrogen atoms of protein and peptide; Position restrain (2 kJ·mol⁻¹·Å⁻²) for the sidechain non-hydrogen atoms of protein and peptide; Planar harmonic restraint (2 kJ·mol⁻¹·Å⁻²) for the phosphorus atom of POPC along the Z-axis; Dihedral restraint (200 kJ·mol⁻¹·rad⁻²) for two dihedrals (C28-C29-C210-C211 and C1-C3-C2-O21).
Step6.5	1 fs	10 ns	Position harmonic restrain (2 kJ·mol⁻¹·Å⁻²) for the backbone non-hydrogen atoms of protein and peptide; Position restrain (0.5 kJ·mol⁻¹·Å⁻²) for the sidechain non-hydrogen atoms of protein and peptide; Planar harmonic restraint (0.4 kJ·mol⁻¹·Å⁻²) for the phosphorus atom of POPC along the Z-axis; Dihedral restraint (100 kJ·mol⁻¹·rad⁻²) for two dihedrals (C28-C29-C210-C211 and C1-C3-C2-O21).
Step6.6	1 fs	10 ns	Position harmonic restrain (0.5 kJ·mol⁻¹·Å⁻²) for the backbone non-hydrogen atoms of protein and peptide;
Step7	2fs	1000 ns	Restrain-free

5. In Supplementary Figure 7. “all snapshots obtained from MD simulations were superimposed on Ca atoms of the compound 2–GLP-1R–Gs complex”. Were all Ca atoms, including both ECD and TMD, used for the protein superimposition?

Response: We thank the reviewer for the comment. In the previous version, we used all Ca atoms for protein superimposition, which is probably not suitable for class B1 GPCRs. Since we performed new MD simulations based on the newly-solved cryo-EM structure of compound 2-bound GLP-1R, all MD snapshots were superimposed on the cryo-EM structure of GLP-1R TMD using the Ca atoms. Both figure and figure legends have been updated to reflect this

change.

6. In Supplementary Figures 7 and 9. Figure 7a compares MD snapshots with the cryo-EM structure. Is the reference structure for calculating Figures 7b Ca RMSD plots also the cryo-EM structure or the frame 0? If not the cryo-EM structure, please indicate the RMSD between the last MD snapshot and the cryo-EM structure.

Response: We thank the reviewer for the comment. Exactly as the reviewer pointed out, the frame 0 was used, where high structure similarity between frame 0 and cryo-EM structure (C α RMSD is 0.2~0.6Å) was observed. To make it clearer, we have updated the RMSD calculation using the cryo-EM structure as reference and revised the figure legends accordingly.

7. In Supplementary Figure 9b. Please explain why the Ca RMSD of the ECD gradually reduces over MD.

Response: We thank the reviewer for the comment. For the RMSD calculation, ECD (residues 29 to 136) and TMD (residues 137 to 423) were independently superimposed on the corresponding segments of the cryo-EM structure of compound 2–GLP-1R–G_s complex using the C α atoms. Thus, Supplementary Figure 9b indicates the high inherent structural stability of ECD in MD simulations compared to the conformation observed in cryo-EM structure. In fact, ECD is in dynamic orientations relative to TMD in our MD simulations (see figure below), consistent with previous reports [Investigation of ECD conformational transition mechanism of GLP-1R by molecular dynamics simulations and Markov state model. *Phys. Chem. Chem. Phys.*, 21, 8470-8481 (2019); Full-length human GLP-1 receptor structure without orthosteric ligands. *Nat Commun* 11, 1272 (2020); Structure and dynamics of semaglutide and taspoglutide-bound GLP-1R–G_s complexes, *bioRxiv*, DOI: 10.1101/2021.01.12.426449]. The figure and its legend have been revised to reflect this point.

RMSD curves of the compound 2–GLP-1–GLP-1R in MD simulations. MD snapshots were first superimposed on the TMD (residues 137 to 423) of the cryo-EM structure using the C α atoms, then subjected to RMSD calculation for ECD and TMD.

Reviewer #3 (Remarks to the Author):

Zhaotong Cong and colleagues present in their manuscript three GLP-1R complex cryo-EM structures in the presence of the ago-allosteric modulator “compound 2”. Their results provide

new structural insights into the agonist and allosteric action of this compound and thereby contribute valuable information towards possible pharmacological applications of such agents. Of special notice is the proposed cooperation of the ECD as an agonist in the absence of orthosteric ligand. The paper will be of great interest among the structural, GPCR, and drug development communities and, in principle, I would recommend it for publication, if the authors can address the following issues:

1. The cryo-EM densities for compound 2 in Suppl. Fig. 3 look quite ambiguous. To provide visual evidence for the statement on line 129 “cryo-EM map demonstrates that compound 2 is covalently bound to C347”, please add a new supplementary figure showing zoomed-in views of the compound and the interacting region of TM6 placed inside the density maps of the three samples.

Response: We thank the reviewer for the comments. Actually, we have prepared the compound 2 alone sample again and resolved the complex structure showing a clear density of compound 2 in the map (see revised Supplementary Fig. 3d below). Thus, we replaced the previous 3.4Å structure with this newly-solved cryo-EM structure of compound 2-bound GLP-1R (2.5Å).

Supplementary Fig. 3d. The zoomed-in views of compound 2 and the interacting region of TM6 placed inside the density maps of compound 2–GLP-1 (left), compound 2–GLP-1–GLP-1R (middle) and compound 2–LY3502970–GLP-1R (right).

2. Line 141, Fig. 2c: A350W and especially K351A seem to not just reduce the response but diminish it almost as much as the C347A mutation. Please rephrase the sentence to reflect the strong effect of these mutations.

Response: This point is well taken. The corresponding sentence has been revised: “The bulky A350^{6.39b}W and K351^{6.40b}A almost abolished the maximal response (E_{max}) of GLP-1R-mediated cAMP accumulation in presence of compound 2”.

3. The presence of another small GLP-1R agonist molecule (WB4-24) in the compound 2-GLP-1R sample is mentioned just in the cryo-EM data acquisition section. At which step was WB4-24 added and in what concentration? It was not mentioned in the complex formation and purification part. Is the binding site of WB4-24 known? If yes, please include a new panel in Suppl. Fig. 11 showing the WB4-24 model imposed on a zoomed-in view of both the global and ECD-resolved maps to reaffirm the absence of density at that location in both maps. Is it possible that WB4-24 contributes to the stabilization of the observed ECD conformation with N-term. helix insertion into the TMD core?

Response: We thank the reviewer for this important concern. We were trying to solve the structure of WB4-24-bound-GLP-1R initially but could not see the density of WB4-24 and ECD in the map (see figure a below). We also prepared a sample by adding compound 2 and WB4-24 together and obtained the complex structure with GLP-1R and G_s (see figure b below). Since we were unable to see the density of WB4-24, we assumed that this was the structure of compound 2-bound GLP-1R as originally presented, without looking into the details of the data obtained from the complex of compound 2-bound GLP-1R in the absence of WB4-24. Following the reviewer's above comment, we analyzed the cryo-EM map and solved the structure of compound 2-bound GLP-1R in complex with G_s . Compared with that originally presented, different conformations of the N-terminal α -helix were noted (see figures c and d below), suggesting the presence of WB4-24 in the cryo-EM sample preparation might have affect the conformation of the ECD. We are presently working diligently to solve the structures associated with WB4-24. To reflect this, the relevant parts of the manuscript including text, figures and supplementary information have been revised with this newly solved, compound 2 only structure.

Effect of WB4-24 during sample preparation on the compound 2–GLP-1R– G_s complex structures. **a**, Complex map of WB4-24 activated GLP-1R (no ECD was seen). **b**, Complex map of compound 2-bound GLP-1R with WB4-24 in the sample preparation. The main difference between them lies in the ECD conformation (**c**, pink with compound 2 only). The N-

terminal α -helix of the compound 2-only structure is closer to ECL1 (**d**).

4. Which map is shown in Fig. 1a? It does not look like either of the maps in Suppl. Fig. 2a.

Response: The 3.4Å map (with the ECD) was shown in Fig.1a. For the reason mentioned above, we used the newly-solved compound 2-bound GLP-1R structure (without WB4-24 during sample preparation) in the revised manuscript with updated Fig. 1a and Suppl. Fig. 2a.

5. Was the compound 2 density present in both the ECD-focused and the global maps? Please deposit both maps in EMDB. Currently, there is only one accession number listed in the data availability section.

Response: The density of compound 2 can be seen both in ECD-focused and global maps. As mentioned above, we applied a new complex structure showing a better compound 2 density and re-uploaded mrc map and PDB files.

6. There are no dataset and processing details for the additional data that was collected for the compound 2-GLP-1R and according to the cryo-EM data acquisition section produced a 3.4Å map. This map was used to confirm similarity with the WB4-24-containing sample map. If the data and map were used in the work, I would strongly encourage including the experimental and processing details in the paper and depositing the additional map in EMDB.

Response: We totally agree with the above comments. For the reason mentioned above, new figures, mrc map and PDB files are used in the revision together with updated experimental and data processing details.

7. It is surprising that the particles used for the ECD-focused map and the particles used for the global map were separate subsets (Suppl. Fig. 2a). Typically, the active state will have the most rigid and well-resolved structure and according to the presented hypothesis, the ECD insertion into the TMD should have further stabilized the global structure. Did the authors try combining the two subsets together to possibly get a better global map and then classifying-out a subset with an ECD-focused mask?

Response: We thank the reviewer for the comments. We agree that the active state usually has the most rigid and well-resolved structure, although conformational dynamics present in many GPCR-G protein complexes, such as PTH1R-G_s (PMID: 30975883) and NTSR1-G_i (PMID: 31243364). We tried to merge the two subsets and reconstructed a map with a quality similar to the one with 2.8Å. Further classification on the ECD region did not result in a better map. This is most likely owing to the limited interactions between the N-terminus of ECD and TMD without a peptidic helical ligand stabilizing the relative orientation of ECD to TMD.

8. Was density for the ECD N-term. helix tip entering the TMD core also present in the current global map (Suppl. Fig. 2a, right)? If not, is the inserted ECD only a transient state that further stimulates G protein association? Please provide a comment on this in the discussion.

Response: We processed the cryo-EM data of the complex with compound 2 only in the same way as before, thus the newly-solved compound 2-bound GLP-1R structure was used to answer this reviewer's question. The conformation of N-terminal helix was consistent in both ECD-containing (3.3Å) and TMD only (2.5Å) maps shown in Suppl. Fig. 2a. Under the lower count level of the volume viewer, the ECD could also be seen that fits well to the one with ECD (see figure below). Therefore, we believe that the insertion of ECD to the TMD core is a unique feature of compound 2-bound GLP-1R. A statement reflecting this observation has been included in the revised discussion: "In line with the cryo-EM structure of compound 2-bound GLP-1R, the MD simulations indicate that the N-terminal helix consistently inserts to the TMD core thereby stabilizing the interaction between the ECD and ECL1. This implies that the insertion of the ECD N-terminal helix to the TMD core is a unique feature of compound 2-bound GLP-1R."

Minor corrections:

- The FSC curves in Suppl. Fig. 2c seem to be misrepresented, FSC_{work} and FSC_{test} in particular. Were the Relion FSC curves for uncorrected/noise-substitutes maps misplaced as FSC_{work} and FSC_{test}?

Response: We thank the reviewer for pointing out our error and the corrected FSC curves (as shown below) have been updated in Suppl. Fig. 2c.

FSC curves of the overall refined structure.

- Line 62: ref.19 does not summarize GLP-1R peptide and small molecule agonist structural

studies, as implied in the sentence.

Response: We thank the reviewer for pointing out our error and reference number 19 was deleted in the revised manuscript.

- Suppl. Fig. 2: please add class distribution values to the workflow graphs

Response: We thank the reviewer for the suggestion and have added class distribution values to the workflow graphs in suppl. Fig. 2.

- Suppl. Fig. 3: for easier interpretation and to avoid confusion, I would suggest presenting the structures in the same order as in Fig. 1 and Suppl. Fig. 2.

Response: We thank the reviewer for the suggestion and the figures were reordered in Suppl. Fig. 2 accordingly.

- Line 143: “evaluated” -> “elevated”

- Line 229: “their overall structure” -> “the overall structure”

- Suppl. Material, line 52: Compassion -> Comparison

- Suppl. Table 1: The electron exposures are listed as 25.3 e/A² while in the methods section it is 80 e/A².

Response: We thank the reviewer for pointing out our errors and have corrected them accordingly in the revised manuscript.

REVIEWERS' COMMENTS

Reviewer #2 (Remarks to the Author):

The authors have sufficiently answered and addressed all of the concerns and are commended for performing/including additional experimental results. Amongst the additional data, the new cryo-EM structure of the compound 2-bound GLP-1R at 2.5Å is very impressive. Congratulations.

A few remaining minor points to consider:

1. There appears to be a typo on p6:150, I think "C348" should be "C438".
2. More clearly clarify/reference "WB4-24".
3. Consider referencing two relevant articles that were reported during the submission/review process: PMID: 33724441 and PMID: 33721487.

-Kyle Sloop

Reviewer #3 (Remarks to the Author):

The authors have addressed the raised concerns and have improved the manuscript substantially. I recommend it for publication with a minor correction in Figure 1a, where the isosurface representation of compound 2 in the 3D map looks very much like it is of the atomic model and not of the cryo-EM map.

Radostin Danev